# Biomaterial-associated molecular patterns (BAMPs) modulate macrophage polarization in bone grafting

Carel Brigi[1,2], Mawieh Hamad[1,3], Ensanya Ali Abou Neel[1,4], Balachandar Selvakumar[1], K. G. Aghila Rani[1], Sausan AlKawas[1,5], Hamza M. Al Hroub[1], Sherlyn Jemimah[6], Amin F. Majdalawieh[6,7], Amjad Mahasneh[6,7], A. R. Samsudin[1,5]*

1 Research Institute of Medical and Health Sciences (RIMHS), University of Sharjah, Sharjah, United Arab Emirates, 2 College of Graduate Studies, University of Sharjah, Sharjah, United Arab Emirates, 3 Department of Medical Laboratory Sciences, College of Health Sciences, University of Sharjah, Sharjah, United Arab Emirates, 4 Department of Preventive and Restorative Dentistry, College of Dental Medicine, University of Sharjah, Sharjah, United Arab Emirates, 5 Oral and Craniofacial Health Sciences Department, College of Dental Medicine, University of Sharjah, Sharjah, United Arab Emirates, 6 Department of Biology, Chemistry and Environmental Sciences, College of Arts and Sciences, American University of Sharjah, Sharjah, United Arab Emirates, 7 Advanced Biosciences and Bioengineering Research Centre, American University of Sharjah, Sharjah, United Arab Emirates

* drabrani@sharjah.ac.ae

## Abstract

Bone graft rejection in oral surgery remains a significant challenge, potentially compromising the success of reconstructive procedures. Biomaterial-associated molecular patterns (BAMPs) play a pivotal role in bone graft integration and rejection by initiating and modulating immune responses during the foreign body reaction (FBR). This study evaluated BAMPs components, physicochemical properties, and adsorbed serum proteomic profiles in demineralized (DMB) and decellularized (DCC) bone grafts. The immunomodulatory effect of adsorbed serum proteins on DMB and DCC to modulate macrophage polarization was also investigated. Serum protein adsorption profiles were determined by incubating 0.5 g of DMB and DCC bone grafts in 10% FBS. Adsorbed serum proteins were buffer-desorbed, quantified, and subjected to proteomic analysis. The capacity of DMB and DCC adsorbed serum proteins to modulate functional and phenotypic profiles of macrophages was assessed using THP-1 cells. Macrophages were incubated for 72 hours with or without DMB- or DCC-adsorbed serum proteins, followed by expression analysis of CD14, CD16, CD86, CD206, HLA-DR, iNOS, Arg-1, IL-1β, TNF-α, IL-10, and TGF-β. Physicochemical analysis revealed that DCC bone grafts' surfaces were hydrophilic, anionic, and smooth, whereas DMB surfaces were hydrophobic, mildly anionic, and rough. The proteomic profiles of adsorbed serum proteins associated with DMB and DCC bone grafts showed significant variations. Macrophages exposed to DCC-adsorbed serum proteins exhibited an elongated cell morphology, increased expression of CD16 and CD206 ($p < 0.0001$), and higher Arg-1 expression. Conversely, the

**Data availability statement:** All relevant data are within the manuscript and its Supporting Information files.

**Funding:** The authors gratefully acknowledge the financial support provided by the University of Sharjah through the targeted grant (Grant No. 2301100169) awarded to ARS. The College of Graduate Studies at the University of Sharjah also generously provided additional research funding under Budget Code 710908, with Reference No. CL2415199, awarded to CB and ARS. The funding agency had no role in the design, conduct, analysis, or reporting of this study.

**Competing interests:** The authors have declared that no competing interests exist.

DMB group macrophages exhibited a rounded cell morphology, increased expression of CD86 and HLA-DR (p < 0.01), and higher iNOS expression. The DCC group showed higher mRNA levels of IL-10 and TGF-β (p < 0.0001). The physicochemical properties of bone grafts govern serum protein adsorption, which, in turn, directs macrophage polarization and graft acceptance. Understanding these interactions guides the design of immunomodulatory biomaterials to enhance graft integration.

## 1. Introduction

Bone grafting techniques address bone defects in oral, maxillofacial, and orthopaedic surgeries caused by tumours, trauma, congenital abnormalities, or degenerative diseases [1,2]. Despite their increasing use for bone regeneration, achieving biological integration and consistent clinical outcomes remains a significant challenge. An adverse host immune response, known as the foreign body reaction (FBR), can render the bone graft ineffective and reduce its clinical performance [3–5]. FBR commences with establishing a provisional matrix, within which a layer of serum proteins adsorbs to the surface of biomaterials. The provisional matrix contains chemoattractants, cytokines, and growth factors that recruit immune cells and initiate inflammation. Immune cells recognize and adhere to the biomaterial surface through the adsorbed protein layer at the cell-biomaterial interface [6]. The acute inflammatory phase following protein adsorption can enhance tissue restoration and osseointegration of bone grafts or cause chronic inflammation, leading to fibrous encapsulation, rejection, and graft failure [7,8]. While various immune cells participate in the biomaterial host response, macrophages are widely recognized as central orchestrators of FBR and play a crucial role in directing the immunological reaction to implanted bone grafts [3,9–11].

Recent studies indicate that macrophages involved in the FBR process can exhibit phenotypic differentiation modulated by proteins derived from cells and serum that are adsorbed on the surface of biomaterials [12,13]. Although these proteins are usually non-inflammatory in interstitial fluid, their adsorption onto a material's surface can cause unfolding and expose epitopes that activate immune cells [12,14–16]. Current literature suggests the hypothesis that macrophage phenotypic modulation in the FBR process is regulated by biomaterial-associated molecular patterns (BAMPs). BAMPs consist of (i) adsorbed serum proteins, (ii) danger signals, and (iii) the biomaterial's physicochemical properties [17–19]. The underlying mechanisms of BAMPs postulate that the surface characteristics of biomaterials influence the adsorbed proteomic profiles and subsequent cellular interactions [20]. BAMPs are relevant to FBR, as the first stage in the FBR process involves protein adsorption to the biomaterial surface, which is determined by the biomaterial's physicochemical properties.

Although previous studies have examined the macrophage response to various bone grafting biomaterials, both in vitro and in vivo settings [21–26], a systematic characterization of the cell-biomaterial interface, including the physicochemical properties of biomaterial, adsorbed serum protein profiles, and macrophage phenotypic

modulation based on adsorbed serum proteins, remains unexplored. Understanding these interconnected factors is crucial for designing bone graft materials that mitigate the FBR and stimulate the immune response to favour bone regenerative outcomes. Such insights are pivotal for developing next-generation bone graft substitutes that can minimize chronic inflammation, promote vascularization, and accelerate functional bone regeneration.

The two types of natural bone scaffolds are demineralized and deproteinized or decellularized bone grafts. Demineralized bone grafts (DMB) undergo processing that removes dense inorganic minerals, such as hydroxyapatite, while preserving the organic matrix, including type I collagen, resulting in a soft, collagen-rich matrix. Conversely, decellularized bone grafts (DCC) processing involves removing all cellular components, such as osteocytes, DNA, and lipids, while maintaining the mineral and collagen structure, resulting in a mineralized, cell-free scaffold that closely resembles native bone [27,28]. The use of two variants of bone grafts derived from a single xenogeneic source in the current study will enable an investigation of how differences in their physicochemical properties will influence protein adsorption and subsequent macrophage differentiation within the host tissue microenvironment. As physicochemical properties and adsorbed serum proteins constitute components of BAMPs, understanding their interactions in regulating macrophage phenotypes will enhance our understanding of the role of BAMPs in FBR to bone grafts.

Based on our previous research, exposure of human peripheral blood monocyte-derived macrophages (PBMMs) to DMB and DCC bone granules resulted in distinct differences in macrophage phenotypic differentiation between the two treatment groups [29]. This observation highlights a crucial knowledge gap in understanding how substrate surface chemistry and bone graft protein adsorption profiles influence macrophage differentiation. Therefore, in the present study, we investigated the physicochemical properties and the adsorbed serum proteomic profiles of DMB and DCC bone grafts and examined the immunomodulatory effects of bone graft adsorbed serum proteins on the phenotypic and functional profile of macrophages.

## 2. Materials and methods

### 2.1. Development of demineralized and decellularized bone grafts

Bone grafts of bovine origin were prepared following established protocols previously published by our research group [28]. Briefly, demineralization involved treating the processed bone samples with 0.6 M hydrochloric acid (HCl). In contrast, decellularization involved a series of chemical treatments, including deproteinization in 4% sodium hypochlorite (NaOCl) and subsequent washes in 0.01%, 0.1%, and 1% sodium dodecyl sulphate (SDS) over 72 hours. SDS-treated scaffolds were incubated in 1% Triton X-100 for 24 hours, then placed in DNase (0.2 mg/mL) and RNase (1 µg/mL) in PBS at 37°C for 1 week. Both DMB and DCC scaffolds were deep-frozen at −80°C for 4 hours, then lyophilized at −40°C under $7 \times 10^{-2}$ millibar pressure. The scaffolds measuring $5 \times 2 \times 2$ mm were packed separately in double-layered plastic pouches and sterilized with gamma radiation at 25 kGy.

### 2.2. Physicochemical characterization of bone grafts

**2.2.1. Surface wettability.** The contact angles between the droplet and the DMB and DCC bone graft surfaces were measured to determine surface wettability. The instrument used for contact angle measurement was an Automated Goniometer (Rame-Hart, Model No. 290-U1). The initially prepared flat bone blocks, measuring $5 \times 2 \times 2$ mm and weighing 0.5g, were secured to the goniometer stage with double-sided adhesive tape. A 7.5 µL droplet of distilled water was gently dispensed onto the bone graft surface. Measurements were taken in triplicate at room temperature (22–23°C), with relative humidity maintained at 50–55% during the analysis. The drop image was captured at set time intervals: 0, 30, 60, 90, and 120 seconds. The surface wettability of DMB and DCC bone grafts was assessed by measuring the water contact angle (WCA) at the set time interval.

**2.2.2. Bone graft particle size and surface charge assessment.** The DMB and DCC processed bone graft scaffolds were coarse-ground using a ceramic mortar and pestle, then finely processed with a high-speed blade mill equipped with a stainless-steel crushing blade and vessel. The bone granules were pulverized for 120 seconds at 8000 rpm and subsequently sieved to produce uniformly sized particles [30]. The final particle size distribution of the bone granules was measured using the Malvern Zetasizer Nano-ZS system (Malvern Instruments, UK), with 0.05 g of bone graft powder added to distilled water in a disposable sizing cuvette. Measurements were conducted at 25 °C with an attenuator setting of 9, a measurement position of 4.65 mm, and a measurement duration of 60 seconds.

The finely powdered bone grafts were measured for zeta potential using the Litesizer™ 500 from Anton Paar. 0.05 g of DMB and DCC bone grafts were added to distilled water in Omega cuvettes with a sample volume of 350 µL. The zeta potential was measured using electrophoretic light scattering (ELS), which evaluates the speed of particles in an electric field. Measurements were obtained using the Kalliope software system. For the zeta potential measurement, the input parameters included a voltage of 200.0 V, a mean intensity of 761.7 k counts/s, and 100 processed runs.

**2.2.3. Surface roughness measurement and elemental composition.** The lyophilized and prepared bone grafts, DMB and DCC, measuring 5 × 2 × 2 mm, were examined for their surface topography and elemental composition using scanning electron microscopy and energy-dispersive X-ray spectroscopy (SEM-EDS). Prior to SEM imaging, the bone grafts were coated with a gold sputter, achieving a thickness of 11.1 nm. SEM images were evaluated using a Thermo Fisher Apreo 2 FE-SEM at magnifications of 60x, 100x, and 500x. The elemental composition was also analysed from three random areas of the scaffold. Subsequently, the ImageJ software was utilized to obtain the surface roughness values for DMB and DCC from the SEM images.

## 2.3. Protein adsorption profiles on DMB and DCC

**2.3.1. Protein adsorption and desorption process from bone grafts.** For protein adsorption, 0.5 grams of DMB and DCC bone grafts measuring 5 × 2 × 2 mm were placed in quadrupoles in a 12-well flat-bottom cell culture plate. Each group contained one control incubated in 2.5 mL PBS and three replicates for protein adsorption assessment in 2.5 mL of 10% fetal bovine serum (FBS) at 37 °C. Protein adsorption was conducted at 1 minute, 30 minutes, 1 hour, 2 hours, 4 hours, and 6 hours to analyse the protein adsorption profile on DMB and DCC, as depicted in S1 Fig. The time interval for protein adsorption was chosen based on previous research into protein adsorption kinetics on biomaterial surfaces [31]. However, the initial and final time points for protein adsorption were chosen to align with the FBR protein adsorption phase. A one-minute point was selected to capture the rapid onset of adsorption, which happens within milliseconds. The 4-hours time point indicates that protein adsorption is critical in the early phase of FBR [19]. To assess the adsorption plateau under the given experimental conditions, the analysis was extended by 2 hours beyond the critical protein adsorption time point, with the final protein adsorption time set at 6 hours.

The protein desorption buffer was prepared using Dulbecco's phosphate-buffered saline (DPBS), with the pH adjusted to 9 by adding sodium hydroxide (NaOH) pellets. Desorption buffer was stored in a borosilicate glass bottle suitable for autoclaving and subsequently sterilized by autoclaving at 121°C and 15 psi (1.05 bar) for 15–20 minutes. Following sterilization, the protein buffer was cooled to room temperature, refrigerated, and stored for protein desorption.

Following the post-incubation protein adsorption period, the desorption buffer was thawed to 37°C, and protein desorption was performed. Bone grafts were carefully held with tweezers and gently rinsed twice with 500 µL of molecular water to remove loosely bound proteins. Adsorbed proteins were then desorbed by placing the bone grafts into a 50 mL Falcon conical centrifuge tube containing 2 mL of desorption buffer. The tubes were subsequently placed in an orbital shaker at 37 °C for 2 hours at 160 rpm to promote protein desorption. The desorbed proteins were filtered through a 0.2 µm sterile filter to eliminate any residual bone particles. The desorbed protein from each group (DMB/DCC), in triplicate, was pooled according to their respective time points and then split into two portions. One portion (100 µL) was used to determine protein concentration via the micro-BCA protein assay, enabling quantification of serum proteins adsorbed to the surfaces

of DMB and DCC at different time points. The remaining portion was stored in a centrifuge tube, immediately frozen, and maintained at −80°C for future cell treatment experiments. The desorbed serum proteins were labelled as demineralized desorbed serum proteins (DMSP) and decellularized desorbed serum proteins (DCSP), desorbed from DMB and DCC bone grafts, respectively. Additionally, mass spectrometry analysis was conducted at the one-hour time point to qualitatively assess the adsorbed serum protein profiles on the surface of DMB and DCC.

**2.3.2. Protein quantification and storage.** Following the manufacturer's instructions, the desorbed serum proteins DMSP and DCSP were quantified using the Pierce™ Bicinchoninic Acid (BCA) assay, and the resulting concentrations were reported in µg/mL. DMSP and DCSP were stored at −80°C until further use for cell culture treatment and proteomic analysis.

**2.3.3. Mass spectroscopic analysis of 10% FBS, DMSP, and DCSP.** The sample preparation for 10% FBS, DMSP, and DCSP was conducted as previously described [32]. In summary, 100 µL of each sample was measured, and the proteins were precipitated by adding 300 µL of methanol (Honeywell, Seelze, Germany). The samples were incubated at −20 °C for 2 hours, then centrifuged at 14,000 rpm for 15 minutes. Subsequently, the protein pellets were air-dried and then resuspended in a denaturation buffer comprising 6 M urea, 2 M thiourea, and 10 mM Tris at pH 8 (100 µL). The protein concentration was determined using the modified Bradford assay (Sigma-Aldrich, St. Louis, MO, USA). For standardization, 100 µg of each sample was reserved to determine the total protein content. The protein digestion and desalting technique was performed as previously described [33]. An ultra-high-pressure liquid chromatography (UHPLC) system from Bruker Daltonics (Bremen, Germany), specifically the Nano-Elute model, was used in conjunction with a Tims-TOF quadrupole-time-of-flight mass spectrometer featuring a Captive Spray ionization source to conduct the LC-MS/MS analysis [32].

The raw data were processed using FASTA libraries, which were uniprotkb_bovine_AND_model_organism, uniprotkb_bovine_serum, and uniprotkb_albumin_bovine. Statistical significance was assessed using ANOVA. To identify serum proteins with significantly higher or lower adsorption on bone graft surfaces, volcano plots were generated, highlighting proteins with fold changes greater or lesser than 2. The observed fold changes in the DMSP and DCSP samples were compared with the control group (10% FBS).

## 2.4. Macrophage treatment response to DMSP and DCSP

**2.4.1. Cell culture protocol.** THP-1 cells were sourced from CLS Cell Line Services GmbH, Eppelheim, Germany, and cultured in RPMI medium supplemented with 10% FBS and 100 U/mL penicillin and 100 µg/mL streptomycin. All reagents were from Sigma-Aldrich, USA. The cells were cultured in a humidified environment with 5% $CO_2$ at 37 °C. THP-1 cells were differentiated into M0 macrophages using a concentration of 80 ng/mL PMA (Sigma-Aldrich, USA) with a cell seeding density of $5 \times 10^5$ cells/mL for 24 hours [34]. After PMA treatment, cells were washed with prewarmed PBS at 37 °C and treated with DMSP for the DMB group and DCSP for the DCC group for 72 hours.

**2.4.2. XTT Assay.** An XTT viability assay was performed to identify the optimal concentration and incubation duration for treating macrophages with adsorbed serum proteins. THP-1 cells were primed into macrophages using 80 ng/mL PMA at a density of $5 \times 10^5$ cells/mL. Macrophages were then seeded into 96-well plates at a density of 50,000 cells per 200 µL per well. After 24 hours of PMA treatment, the macrophages were washed with pre-warmed PBS. Macrophages were then treated with 200 µL of DMSP and DCSP at concentrations of 25, 50, and 100 µg/mL [35] for the DMB and DCC groups, respectively. The cells were incubated for 24, 48, and 72 hours. Following these incubation periods, the cells were treated with 50 µl of XTT solution (Cell Proliferation Kit II (XTT), Sigma-Aldrich, USA), prepared according to the manufacturer's guidelines. Subsequently, the macrophages were incubated for an additional four hours. The absorbance was then quantified using a plate reader at 490 nm with a 1.0-second exposure time per well.

**2.4.3. Macrophage treatment protocol.** THP-1 cells were differentiated into macrophages with PMA at 80 ng/mL and seeded at $5 \times 10^5$ cells/mL per well [34] in a 6-well plate. After 24 hours, the macrophages were washed with pre-warmed

PBS and divided into five groups. For the DMB treatment group, the treatment solution was prepared using DMSP collected at the 1-hour time point (stock concentration ≈325 µg/mL), stored at −80 °C, and thawed at 37 °C. The thawed DMSP was then diluted in RPMI medium supplemented with 1% penicillin–streptomycin to achieve a final concentration of 50 µg/mL. Macrophages were then treated with 3 mL of the DMSP treatment solution for 72 hours per well. Similarly, for the DCC treatment group, macrophages were exposed to 3 mL of DCSP-containing medium at a final concentration of 50 µg/mL, prepared from a 1-hour DCSP stock solution (≈285 µg/mL). Macrophages in the negative control group of each well were treated with 3 mL of RPMI medium supplemented with 1% Pen-Strep and 500 µL of protein desorption buffer. A volume of 500 µL of desorption buffer was used in the negative control, as this was the approximate volume of DMSP or DCSP taken from the stock required to reach the target working concentration in the test group. The two positive control groups, LPS and IL-4, were used to activate macrophages at concentrations of 100 ng/mL and 15 ng/mL, respectively, promoting both pro-inflammatory and anti-inflammatory responses. The cell treatment protocol for the five groups is shown in S2 Fig.

**2.4.4. Cell surface morphology.** THP-1-derived macrophages were seeded at a density of $5 \times 10^5$ cells/mL on a 25 mm sterile coverslip. The treatment procedure for the control, DMB, and DCC groups was carried out as previously described. After 72 hours of exposure to adsorbed serum proteins, cell morphology was examined using SEM. Following PBS wash, cells were fixed on a coverslip with 2.5% glutaraldehyde for one hour and rinsed with deionized water. Serial dehydration was performed using ethanol concentrations of 25%, 50%, 75%, 95%, and 100%, with each step lasting 5 minutes. Finally, specimens were gold sputter-coated by Quorum Technologies SC7620 before being imaged with a Thermo Fisher Apreo 2 FE-SEM.

**2.4.5. Confocal Immunofluorescence Analysis.** THP-1-derived macrophages were seeded at a density of $5 \times 10^5$ cells onto a 25 mm sterile coverslip. Macrophages were treated in the controls and test groups as previously mentioned. After 72 hours post-treatment, the coverslip was washed three times with PBS and processed for confocal immunofluorescence. Cells were fixed in 4% paraformaldehyde (Sigma-Aldrich, St. Louis, MO, USA) for 20 min at room temperature, then permeabilized with 0.1% Triton X-100 (Sigma-Aldrich, St. Louis, MO, USA) in PBS for 5 min. Macrophages were immunostained with Phycoerythrin (PE)-conjugated anti-iNOS antibody for inducible nitric oxide synthase (iNOS) expression (NBP2−22119PE, Novus). Arginase-1 (Arg-1) expression was evaluated using an anti-Arg-1 primary antibody (BioLegend, Cat: 678802), followed by detection with Goat anti-mouse IgG H&L secondary antibody conjugated with Alexa Fluor® 488 (ab150113, Abcam). Nuclear counterstaining utilized 4′,6-diamidino-2-phenylindole (DAPI) (Abcam, Cambridge, UK). Fluorescent signals were visualized using a confocal laser scanning microscope (CLSM; Nikon Eclipse Ti-S, Nikon Instruments Inc., Melville, NY, USA) at 60x magnification.

**2.4.6. Flow cytometry.** The phenotypic assessment of macrophages was evaluated based on cell surface markers, including CD11b, CD14, CD16, CD86, CD206, and HLA-DR. The specific fluorochromes for each antibody are detailed in the accompanying S1 Table. The cell staining procedures and protocol were used as previously published [29]. Following incubation, the cells were washed once and resuspended in 250 µL of fluorescence-activated cell sorting (FACS) buffer. The phenotypic evaluation of macrophages was performed using a FACS Aria flow cytometer, and the resulting data were analysed with FlowJo v10 software (BD, United States).

**2.4.7. qRT-PCR.** After 72 hours of macrophage treatment, RNA was extracted from the samples using the Qiagen RNA Minikit (USA). RNA concentration was measured using a NanoDrop ND-1000 (Thermo Scientific, USA). Subsequently, cDNA was synthesized using reverse transcriptase and a cDNA synthesis kit (Qiagen). Gene expression levels were quantified by real-time RT-PCR using 5X FIREPOL SYBR Green master mix (Solisbiodyne, USA), and GAPDH was used as an internal control. The gene-specific primers used in this research are outlined in Table 1.

**2.4.8. ELISA.** Cell-culture supernatants from each treatment were collected to quantify cytokine levels using ELISA kits. The concentrations of cytokines IL-1β, TNF-α, IL-10, and TGF-β were assessed following the ELISA protocol provided by the manufacturers. Details regarding the ELISA kits for each cytokine are outlined in S2 Table.

**Table 1. Primer sequences used for qRT-PCR in phenotypic assessment of macrophages.**

| Genes | Primer Sequence |
|---|---|
| GAPDH | Forward primer: 5′ -CCACTCCTCCACCTTTGACG- 3′<br>Reverse primer: 5′ -CCACCACCCTGTTGCTGTAG- 3′ |
| TGFβ | Forward primer: 5′ - GACTTCAGCCTGGACAACGA-3′<br>Reverse Primer: 5′-TGTAGGGGTAGGAGAAGCCC-3′ |
| IL-10 | Forward primer: 5′ - GGTCGTGTGCTTGGAGGAAG- 3′<br>Reverse Primer: 5′ - AGCAGGTGACTCCCACTGTA- 3′ |
| TNF-α | Forward primer: 5′ - CAAGGACAGCAGAGGACCAG- 3′<br>Reverse Primer: 5′ - TCCTTTCCAGGGGAGAGAGG- 3′ |
| IL1-β | Forward primer: 5′ - AACCTCTTCGAGGCACAAGG- 3′<br>Reverse Primer: 5′ - AGCCATCATTTCACTGGCGA- 3′ |

## 2.5. Statistical analysis

One-way ANOVA and Student's t-test were performed using GraphPad Prism 10. One-way ANOVA was utilized for data analysis, followed by Bonferroni's multiple comparison test. A p-value of less than 0.05 was considered statistically significant. All data are presented as the SEM (Standard Error of Mean) of at least three independent experiments.

## 3. Results

### 3.1. DMB bone grafts exhibited higher WCA values compared to DCC

At 0 seconds, DMB exhibited a higher WCA value of 103°, while DCC recorded a WCA of 83.5°. At 120 seconds, DMB's WCA decreased to 77°, whereas DCC's WCA dropped further to 44.3°. Fig 1A presents the graphical representation of the WCA for both DMB and DCC from 0 to 120-second interval, indicating a significantly higher WCA for DMB than DCC ($p < 0.001$). Table 2 presents contact angle measurements for DMB and DCC from 0 to 120 seconds. These findings suggest that DMB is hydrophobic while DCC is hydrophilic.

### 3.2. DMB and DCC bone grafts exhibited significantly different surface charge values

The nominal particle size of DMB and DCC bone grafts for zeta potential measurement was 373 nm±44 and 382 nm±71.07, respectively. The zeta potential measurements shown in Figs 1B and 1C for DMB and DCC in physiological saline at pH 7 were −12±0.3 mV and −23.1±0.9 mV, respectively. This demonstrates that DCC has a more negatively charged surface than DMB. The peak zeta potentials were −23 mV for DCC and −12 mV for DMB, with narrow, symmetric distributions, indicating consistent particle size and surface charge. The relative frequency for DMB was slightly higher, at approximately 3.5%, while that for DCC was around 2.8%, indicating that the DMB sample has a marginally more uniform particle population than the DCC sample.

### 3.3. DCC bone grafts had smoother surfaces relative to DMB bone grafts

The SEM analysis revealed distinct surface topographies between the two bone grafts at magnifications of 60x, 100x, and 500x (Fig 2). The DMB group exhibited a porous structure with interconnected macro- and micropores and a rough surface texture. Conversely, the DCC group presented a smoother texture with rounded macropores and micropores. The EDS analysis of the DMB surface indicated a high carbon content of 80%, likely resulting from the HCL demineralization treatment during the bone graft process in Fig 3A. This treatment produced an etched surface with reduced levels of calcium and phosphorus ions. The calcium and phosphorus percentages on DMB bone graft surfaces were 2.1% and 1.9%, respectively. In contrast, the EDS analysis of the DCC surface showed a composition rich in calcium and phosphate,

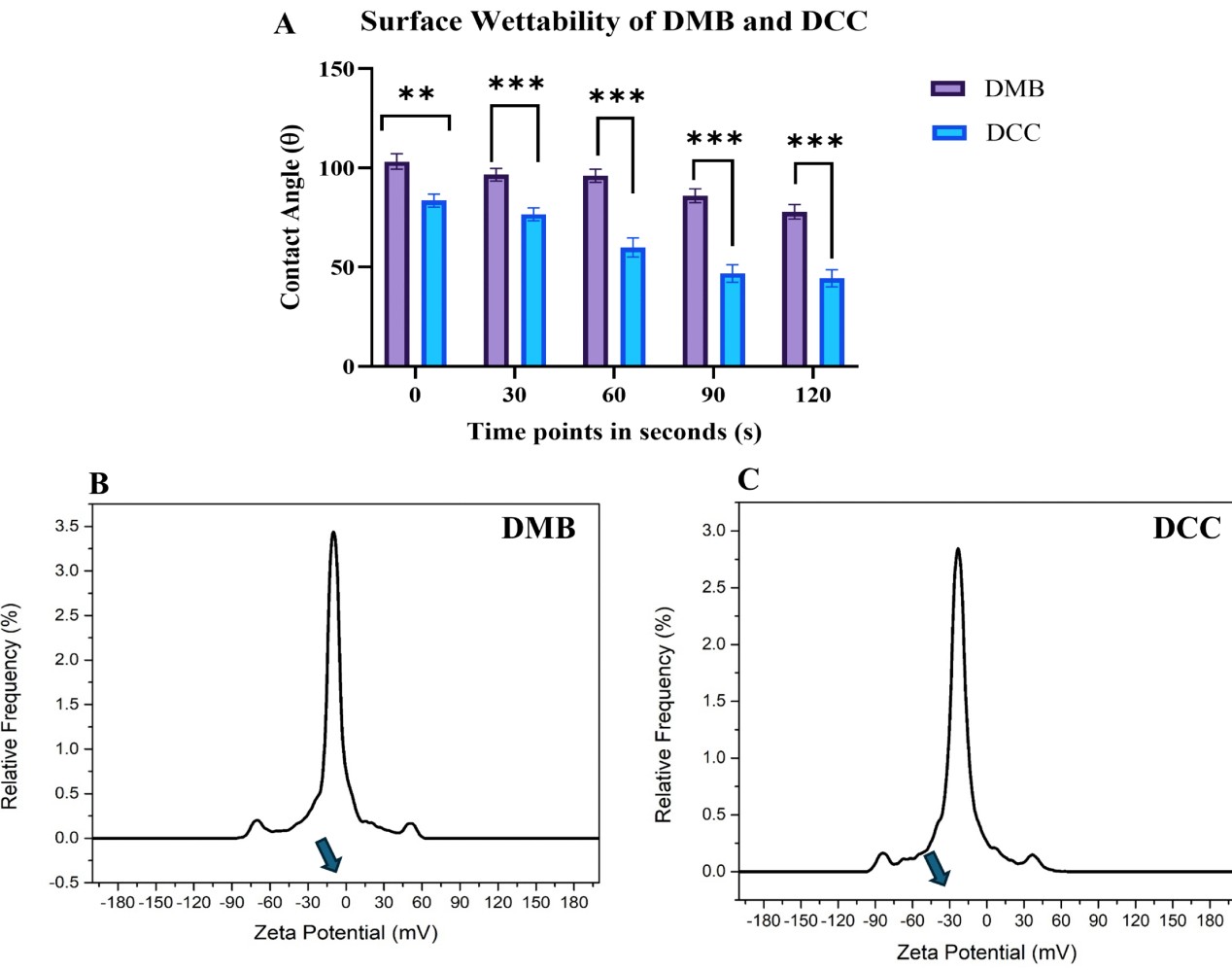

Fig 1. **Surface wettability and charge of DMB and DCC bone grafts.** (A) Graphical depiction of WCA for DMB and DCC indicates a reduced WCA for DCC across time points from 0 to 120 seconds. **(B&C)** The zeta potential measurements indicate values of −12 mV for DMB and −23 mV for DCC.

Table 2. **WCA values for DMB and DCC at 0 to 120 seconds with a standard deviation.**

| Time points in seconds (s) | DMB | DCC |
| --- | --- | --- |
| 0 | 103.20±3.9 | 83.50±3.1** |
| 30 | 97.51±3.2 | 76.55±3.5*** |
| 60 | 95.05±2.9 | 59.85±4.8*** |
| 90 | 85.95±3.5 | 46.75±4.4*** |
| 120 | 77.85±3.7 | 44.35±4.3*** |

Data presented as mean±SDs are based on three independent experiments.***p ≤ 0.001, **p ≤ 0.01.

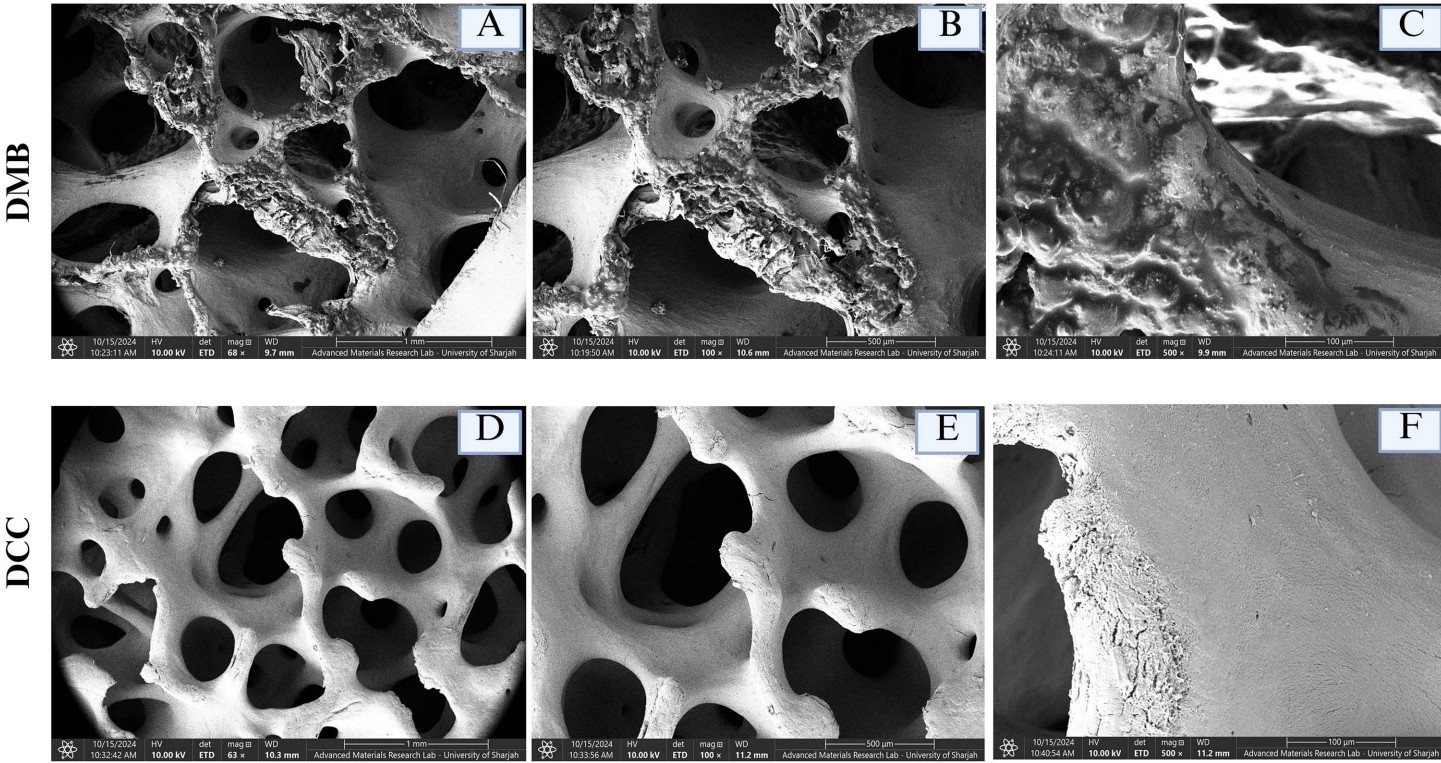

**Fig 2. SEM micrographs of DMB and DCC.** SEM micrographs at 60×, 100×, and 500× magnifications for DMB and DCC reveal distinct surface morphologies. The DMB samples (A-C) exhibit a highly porous architecture with a rough surface topography, whereas the DCC samples (D-F) display a comparatively smooth surface texture. The scale bars of SEM images are as follows: (A, D) 1 mm; (B, E) 500µm; (C, F) 100µm.

with 31.2% and 13.9%, respectively (Fig 3B). Additionally, the DCC surface revealed trace amounts of Mg (0.3%) and Na (0.8%). The gold (Au) peaks corresponded to the conductive sputter coating before SEM imaging.

The 3D surface plots for DMB and DCC were generated in ImageJ, as shown in Figs 3C and 3D, respectively. The DMB exhibited pronounced peaks and valleys, with surface roughness values of Ra at 8.6±0.43 µm and Rz at 5.1±0.6 µm. In contrast, the DCC showed less height variation and a more consistent surface, with Ra and Rz values of 3.6±0.75 µm and 2.3±0.52 µm, respectively (Table 3). These results demonstrate that DMB had higher surface roughness than DCC.

### 3.4. DMB bone grafts demonstrated higher protein adsorption in comparison to DCC

DMB and DCC bone grafts showed a progressive increase in serum protein adsorption over time, with DMSP consistently higher than DCSP at all points (Fig 4A). The control DMSP and DCSP graphs remained near zero throughout, confirming that no proteins were released from the bone grafts in the control groups. The error bars reflect inherent variability in the physicochemical properties of the xenografts, even when obtained from a single source; nevertheless, the overall trends remain statistically distinct. Protein adsorption on DMB bone grafts was consistently higher than on DCC across time points from 1 min to 6 hours. DMB and DCC surfaces demonstrated a rapid escalation in protein adsorption during the initial one hour, followed by a slower rate of increase (plateauing) between one and six hours. Therefore, the protein adsorption profiles on DMSP and DCSP at the one-hour time point were utilized for proteomic analysis and cell treatment.

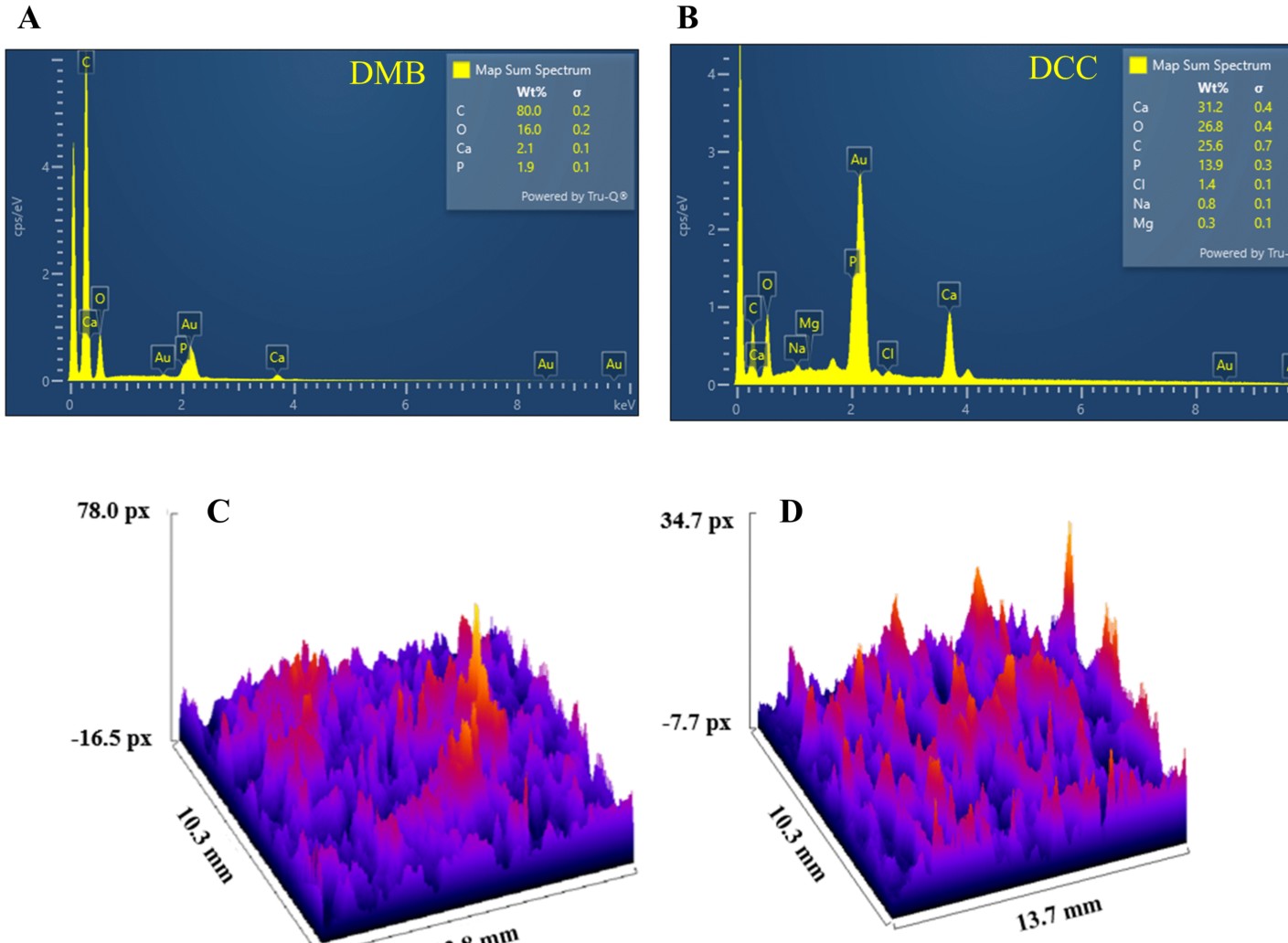

**Fig 3. EDS analysis and 3D surface topography plots of DMB and DCC bone grafts. (A)** EDS analysis of DMB indicates a carbon-rich surface, accompanied by small amounts of calcium and phosphorus. **(B)** EDS analysis of DCC demonstrates a surface abundant in calcium and phosphate, with trace amounts of magnesium and sodium. **(C& D)** 3D surface topography plots of DMB and DCC samples show surface roughness over a 10.3 mm × 13.8 mm area. ImageJ generated the 3D plot in pixels, highlighting peaks and valleys based on the SEM 2D surface image's grayscale.

**Table 3. Represents the surface roughness values of DMB and DCC, detailing their Ra and Rz measurements.**

| Bone Grafts | Ra (µm) | Rz (µm) |
|---|---|---|
| DMB | **8.6±0.43 | ***5.1±0.6 |
| DCC | 3.6±0.75 | 2.3±0.52 |

DMB exhibits higher Ra and Rz values than DCC. The data are presented as mean±SD from three independent experiments. ***$p \leq 0.001$, **$p \leq 0.01$.

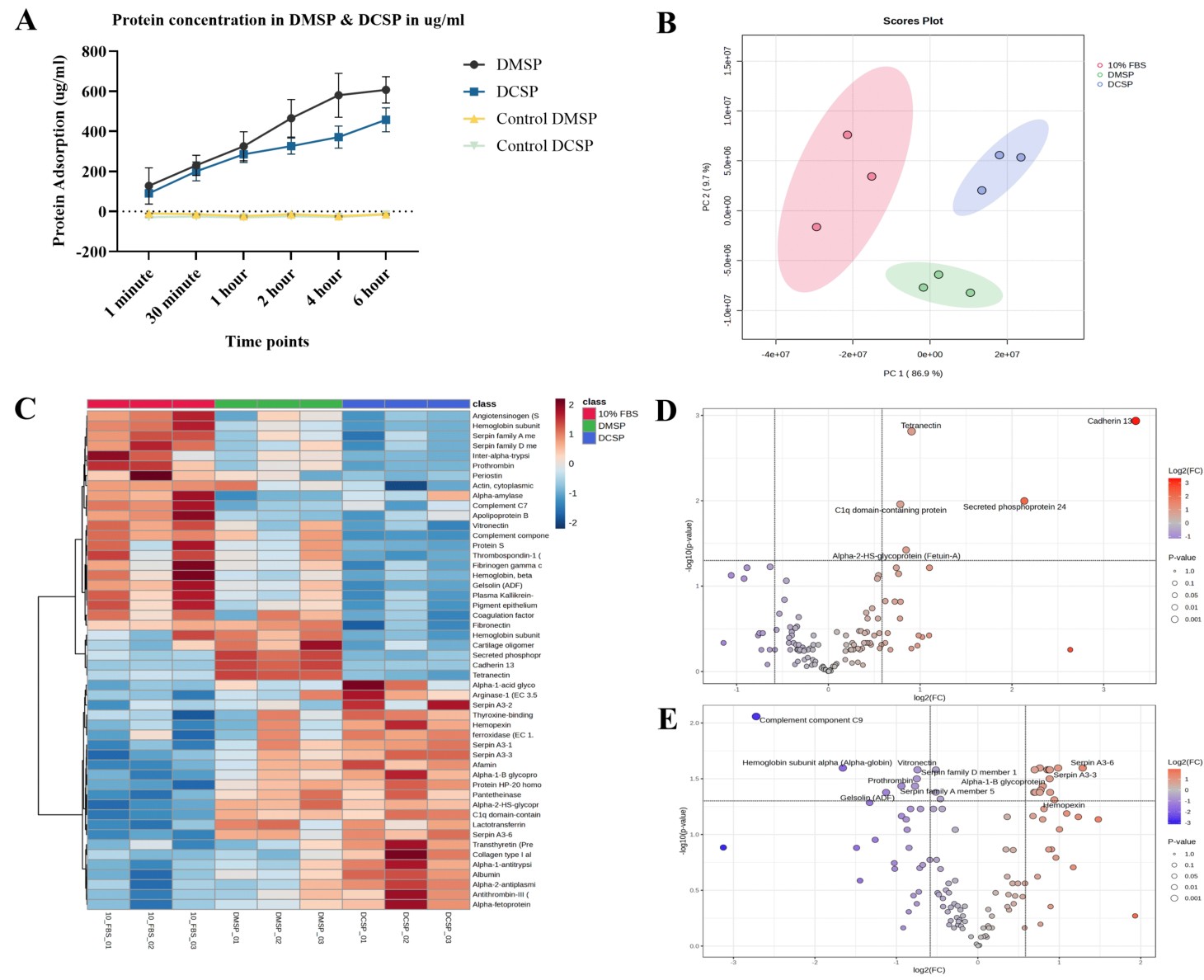

**Fig 4. Serum protein adsorption on DMB and DCC. (A)** Time-dependent adsorption shows higher serum protein adsorption on DMB (labeled DMSP) than on DCC (labeled DCSP). Protein concentration (µg/mL) in DMSP (black) and DCSP (blue) is measured from 1 minute to 6 hours. Control groups (yellow and green) represent baseline values for bone grafts exposed to 2.5 ml PBS for the same duration. **(B)** PCA-based clustering of protein profiles associated with DMSP, DCSP, and 10% FBS. **(C)** The heat map shows hierarchical clustering of protein profiles across 10% FBS, DMSP, and DCSP. **(D)** Volcano plot comparing the proteomic profiles between DMSP and 10% FBS. **(E)** Volcano plot comparing the proteomic profiles between DCSP and 10% FBS, highlighting significantly higher adsorption in red and lower in blue.

## 3.5. Distinct proteomic profiles associated with adsorbed serum proteins on DMB versus DCC

Principal component analysis was performed to examine the variation in proteomic profiles among samples with 10% FBS, DMSP, and DCSP. As shown in Fig 4B, PC1 explains 86.9% of the total variance, and PC2 accounts for 9.7%. This indicates that the profiles of serum proteins adsorbed to DMB and DCC differ from those in 10% FBS. The close clustering

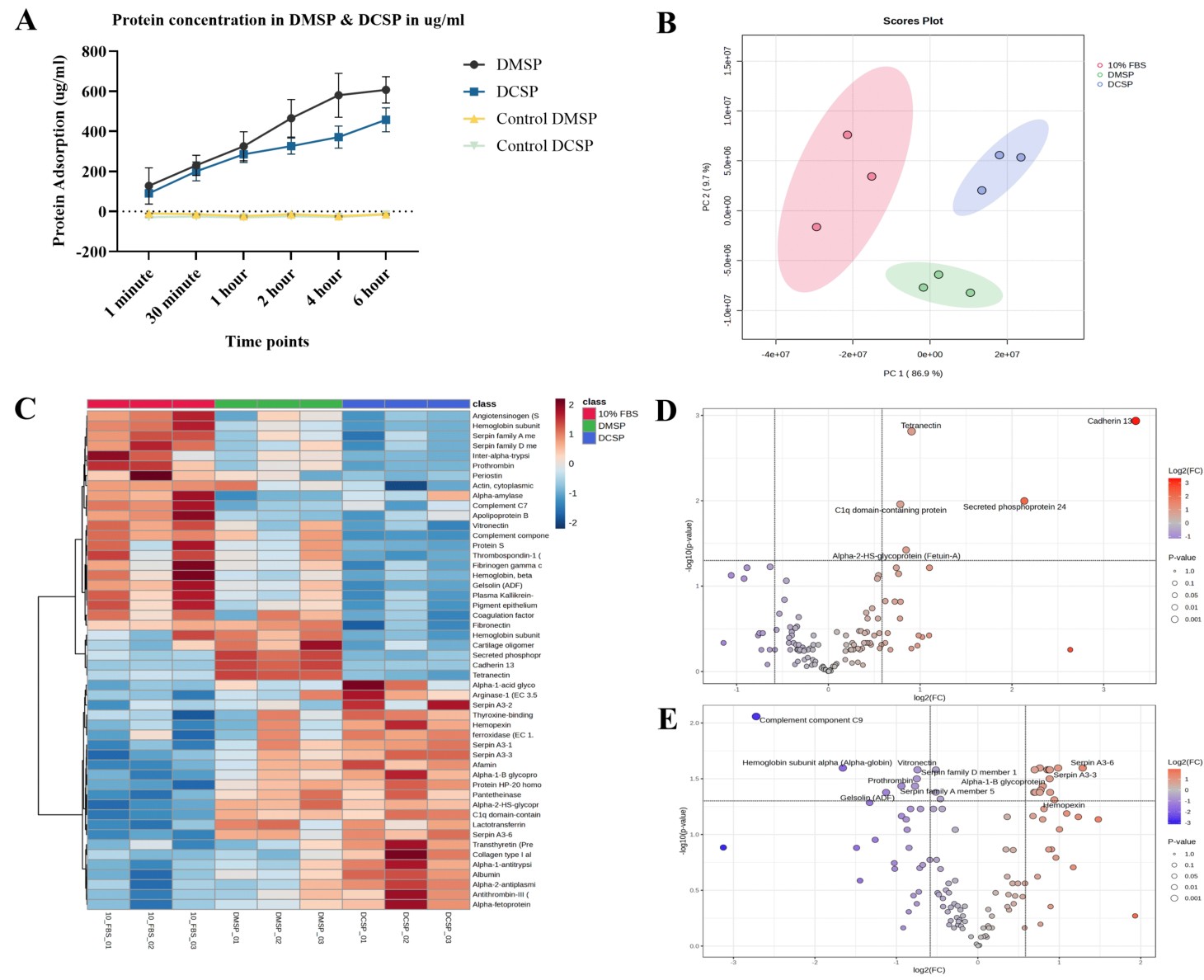

observed within each group, combined with the minimal overlap between groups, indicates a high degree of intra-group consistency and notable inter-group differences in protein adsorption pattern.

Heatmap displays hierarchical clustering of proteomic data for the top 50 quantified proteins in the 10% FBS, DMSP, and DCSP bone grafts in Fig 4C. The colour gradient indicates normalized expression levels, with red representing elevated expression and blue indicating reduced expression. The distinct clustering patterns were apparent, with the 10% FBS group reflecting the various serum protein components it contains. In contrast, the DMSP and DCSP groups exhibited serum proteins present in 10% FBS at varying concentrations being adsorbed onto the DMB and DCC bone graft surfaces. The various serum proteins (albumin, angiotensinogen, prothrombin, plasma kallikrein, serpin A3) and extracellular matrix proteins (periostin, fibronectin, vitronectin, cadherin-13, tetranectin) were differentially adsorbed on the surfaces of DMB and DCC, indicating their varied serum protein concentration in DMSP and DCSP, respectively.

The volcano plots in Figs 4D and 4E illustrate the differential protein adsorption on DMB and DCC compared to 10% FBS, respectively. Statistically significant changes are indicated by points beyond the dotted threshold lines. The protein adsorption profile on DMB, represented as DMSP, when compared to 10% FBS, as illustrated in Fig 4D, demonstrates an upregulation of serum proteins, including cadherin 13, secreted phosphoprotein 24, tetranectin, C1q domain-containing protein, and alpha-2-HS-glycoprotein (Fetuin-A), with a $\log_2 FC > 1$ and a significance level of $p < 0.01$. The differential protein adsorption observed on DCC bone grafts (DCSP) compared with 10% FBS is shown in Fig 4E. It indicated the serum proteins that showed significant downregulation, including complement component C9, hemoglobin subunit alpha, vitronectin, serpin family D member 1, alpha-1B glycoprotein, and gelsolin, all of which showed reduced adsorption to the DCC surface. Conversely, serpin A3-3, serpin A3-6, and hemopexin were significantly upregulated and demonstrated a high level of adsorption on the surface of DCC.

### 3.6. DMB showed higher cell viability than DCC on the XTT assay

The XTT analysis in Fig 5 demonstrated increased cell viability for the DMB and DCC groups compared to the control groups. The DMB group exhibited a time-dependent increase in cell viability, indicating sustained proliferative or metabolic activity over 72 hours during which macrophages were treated with DMSP. Similarly, the DCC group showed increased cell viability relative to the control group following treatment with DCSP, although this was reduced compared to the DMB group. Both groups showed improved cell viability from 24 to 72 hours, with the DMB group exhibiting higher levels. This observation suggests that the desorbed serum proteins from DMB and DCC bone grafts, specifically DMSP and DCSP, respectively, promote cell proliferation over time, with the DMB group potentially demonstrating greater effectiveness. Since the minimum treatment concentration of 50 µg/mL showed a significant difference in cell proliferation between the DMB and DCC groups at all time points, this concentration was chosen for subsequent cell treatment to assess macrophage phenotypic differentiation.

### 3.7. Macrophages showed varied morphology in DMB and DCC groups on SEM

The SEM analysis of macrophage morphology, shown in Fig 6, revealed that the control group of macrophages treated with PBS (pH 9) for 72 hours had an adherent, flattened shape with ruffled edges. In contrast, the DMB group macrophages exhibited a rounded morphology at 2500x and 5000x magnification after treatment with 50 µg/mL of DMSP. The macrophages in the DCC group, on the other hand, had an elongated morphology and showed clearer surface markers after treatment with 50 µg/mL DCSP.

### 3.8. Elevated iNOS expression in the DMB group and increased Arg-1 levels in the DCC group

Immunofluorescence analysis of the metabolic enzymes iNOS and Arg-1 in THP-1-derived macrophages across the control, LPS, IL-4, DMB, and DCC groups is shown in Figs 7A and 7B. In the DMB group, macrophages treated with DMSP exhibited elevated immunofluorescence for iNOS labelled with PE, indicating an increased presence of M1 macrophage subsets that produce higher levels of nitric oxide synthase, as shown in Fig 7A. Conversely, macrophages in the DCC

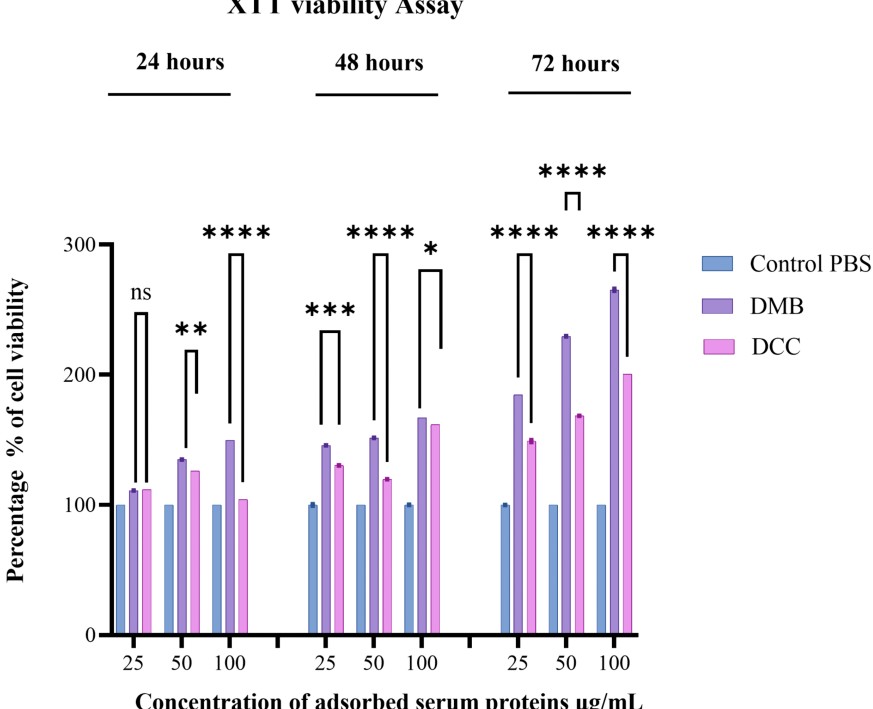

**Fig 5. Cell viability of macrophages following treatment in the control, DMB, and DCC groups.** Assessed over time at 24, 48, and 72 hours. The notation ns indicates no significance, while ****$p \le 0.0001$, ***$p \le 0.001$, **$p \le 0.01$, and *$p \le 0.05$.

group treated with DCSP demonstrated significantly greater immunofluorescence for Arg-1 stained with AlexaFluor-488 when compared to both the DMB and control groups, as illustrated in Fig 7B. The DMB group mirrored the LPS treatment, whereas the DCC group was similar to the IL-4 group. This finding indicates that M2 or anti-inflammatory macrophage phenotypes, characterized by arginase expression, are more prevalent in the DCC group than in the control or DMB groups. Meanwhile, the DMB group has a higher proportion of M1 or pro-inflammatory subsets, which exhibit higher iNOS expression. Figs 7C and 7D depict the graphical representation of the percentage of iNOS-positive and arginase-1-positive macrophages within the test and control groups.

### 3.9. DMB showed higher proinflammatory, and DCC exhibited higher anti-inflammatory mRNA cytokine levels

The results of gene expression analysis for the pro-inflammatory cytokines IL-1β and TNF-α, as well as the anti-inflammatory cytokines IL-10 and TGF-β, are presented in Figs 8A-D. The DMB group showed higher levels of IL-1β and TNF-α expression at $p < 0.05$, like the pro-inflammatory control, LPS. In contrast, the DCC group showed increased expression of IL-10 and TGF-β at $p < 0.0001$ and $p < 0.0001$, respectively, consistent with the anti-inflammatory control, IL-4. These findings support our immunofluorescence analysis, indicating greater pro-inflammatory macrophage presence in the DMB group treated with DMSP and elevated anti-inflammatory macrophage presence in the DCC group treated with DCSP.

### 3.10. ELISA revealed higher pro-inflammatory cytokines in DMB and increased anti-inflammatory cytokines in DCC

The expression of cytokines IL-1β, TNF-α, IL-10, and TGF-β by macrophages was assessed in culture supernatants from the negative control (PBS), positive controls (LPS and IL-4), and test groups (DMB and DCC) at 72 hours, as shown in

| Control | DMB | DCC |
|---|---|---|

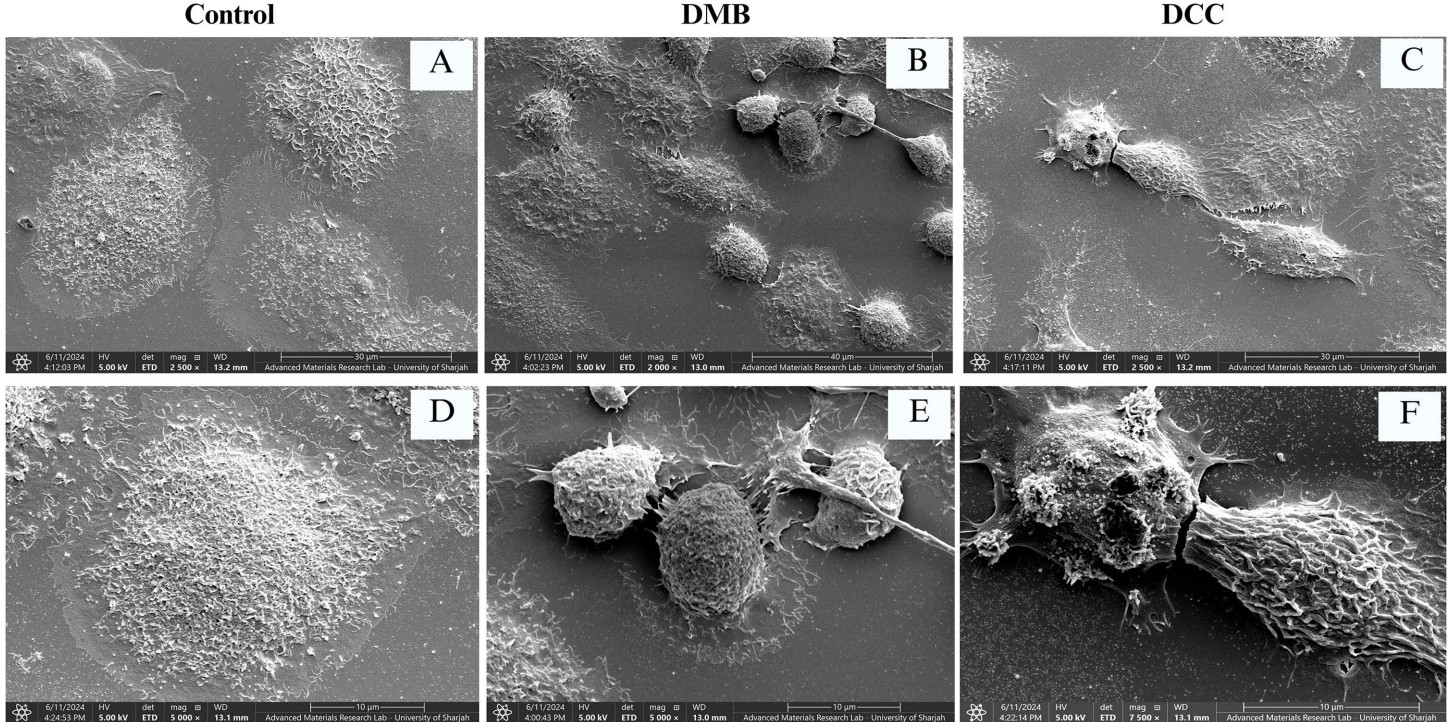

**Fig 6. Cell morphology of macrophages following treatment in the control, DMB, and DCC groups.** SEM images show the macrophage cell morphology on a coverslip after 72 hours of protein treatment in control, DMB, and DCC groups. The scale bars of SEM images are as follows: (A, C) 30μm; (B) 40μm; (D, E, F) 10μm.

Figs 8E- H. The DMB group demonstrated significantly higher expression of the pro-inflammatory cytokines IL-1β and TNF-α than the DCC group, consistent with the positive control, LPS. Furthermore, the anti-inflammatory cytokines TGF-β and IL-10 were elevated in the DCC group relative to the DMB group, with the DCC exhibiting levels comparable to those of the positive control, IL-4.

### 3.11. DMB showed elevated M1 markers, and the DCC group had increased M2 markers on FACS

The phenotypic differentiation of macrophages, following treatment with DMSP and DCSP, was assessed for the expression levels of CD14, CD16, HLA-DR, CD86, and CD206 through FACS analysis in Fig 9. The gating strategy used to identify the macrophage population is presented in S3 Fig. The flow cytometry dot plot depicted in Fig. 9A illustrates the CD14$^+$/CD16$^+$ populations across all five groups in quadrants Q1 and Q3. An increased expression of CD14$^+$ macrophages was observed in the LPS group. Additionally, a higher prevalence of CD16$^+$ macrophages was observed in the DCC group (p < 0.0001), which correlated with the IL-4 group. The quadrant dot plot in Fig 9B presents the CD86$^+$/CD206$^+$ macrophage population in quadrants Q1 and Q3. A higher expression of CD86$^+$ macrophages was observed in the DMB group, mirroring LPS; conversely, the DCC group exhibited increased expression of CD206, paralleling the IL-4 group. The expression of HLA-DR was significantly pronounced in the DMB group at p < 0.001 compared to DCC in Fig 9C. In contrast, the DCC group demonstrated a significantly lower population of HLA-DR$^+$ macrophages than the DMB group (p < 0.001). The FACS analysis revealed a higher proportion of M1 macrophages in the DMB group and a greater predominance of M2 (anti-inflammatory) macrophages in the DCC group (Fig 9C).

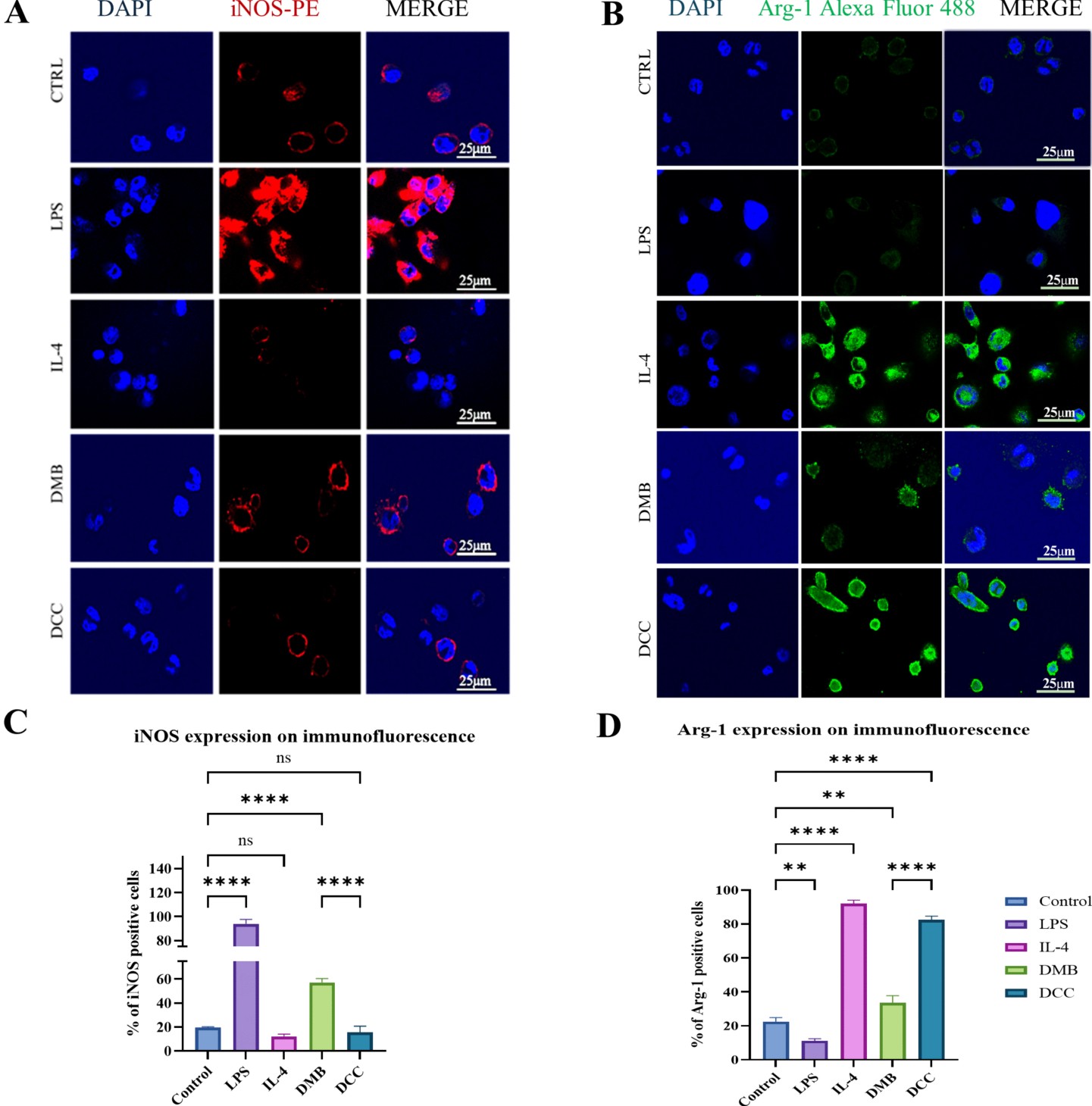

**Fig 7. Metabolic enzyme expression in macrophages following treatment in the control, LPS, IL-4, DMB, and DCC groups. (A)** Representative confocal microscopy images demonstrate iNOS expression in macrophages post-treatment with control PBS, LPS, IL-4, DMB at 50 µg/mL of DMSP, and the DCC group with DCSP at 50 µg/mL. Nuclei are stained with DAPI (blue), while iNOS displays a red fluorescence signal labelled with PE fluorochrome as the intracellular marker. The merged images of DAPI and iNOS-PE include a 25 µm scale bar. **(B)** Confocal microscopy images exhibit the immuno-fluorescence for the macrophage marker Arg-1. Nuclei are stained with DAPI (blue), and Arg-1 is marked by a green fluorescence signal labelled with

the Alexa Fluor 488 fluorochrome as the intracellular marker. (C) and (D) represent the graphical presentation of the percentage of iNOS-positive and arginase-1-positive cells, respectively. The notation ns indicates no significance, while ****p ≤ 0.0001, **p ≤ 0.01.

## (A) – (D) mRNA expression of cytokines (qRT-PCR)

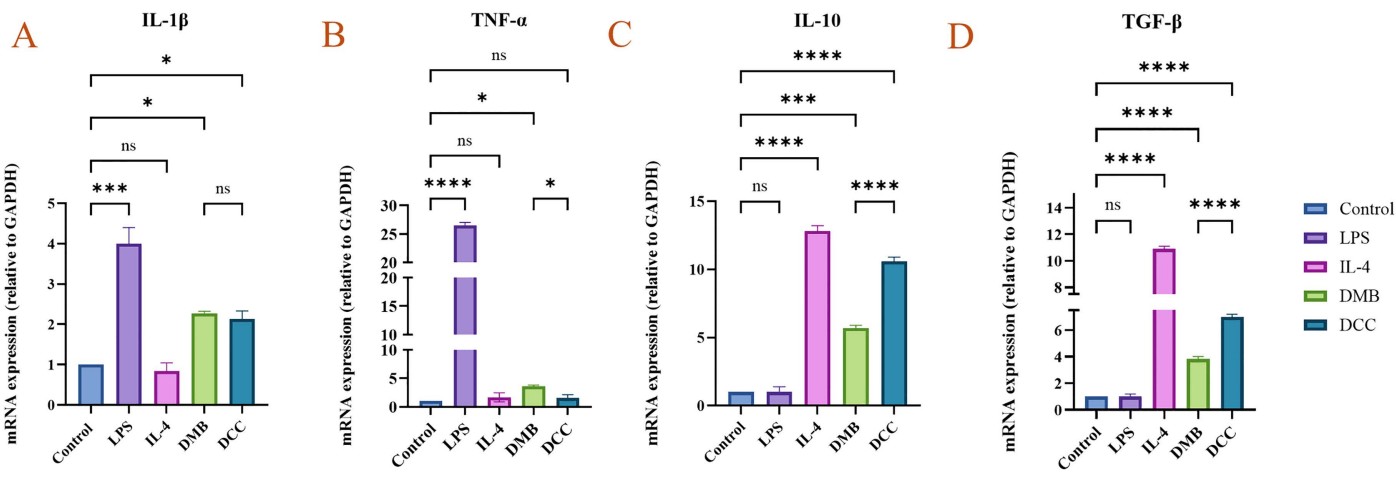

## (E) – (H) Protein concentration of cytokines (ELISA)

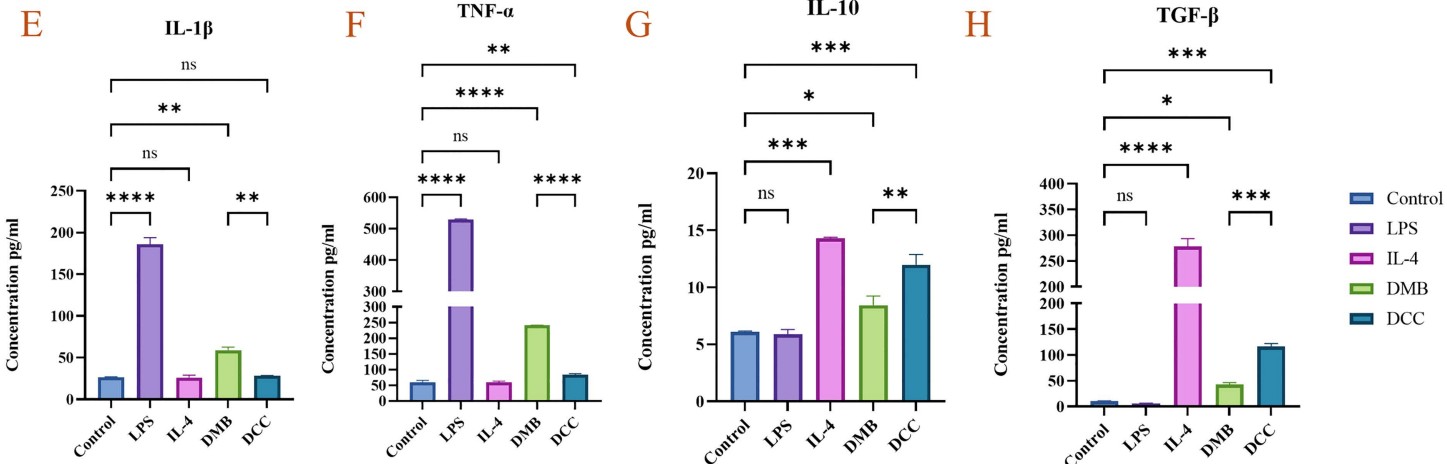

**Fig 8. Illustrates the mRNA and protein expression of pro-inflammatory and anti-inflammatory cytokines.** (A) to (D) indicates the mRNA expression levels of IL-1β, TNF-α, IL-10, and TGF-β relative to the housekeeping gene GAPDH. (E) to (H) shows the protein expression levels of IL-1β, TNF-α, IL-10, and TGF-β in pg/mL, as measured by sandwich ELISA in cell culture supernatant. The notation ns indicates no significance, while ****p ≤ 0.0001, ***p ≤ 0.001, **p ≤ 0.01, and *p ≤ 0.05 represent varying levels of statistical significance.

## 4. Discussion

The serum proteins that adsorb to the surfaces of bone grafts or biomaterials are considered the initial step in FBR, and the formation of these adsorbed serum proteins is seen as the primary mechanism by which macrophages identify foreign bodies [36]. Therefore, it is crucial to understand the surface characteristics and serum proteomic profiles adsorbed on

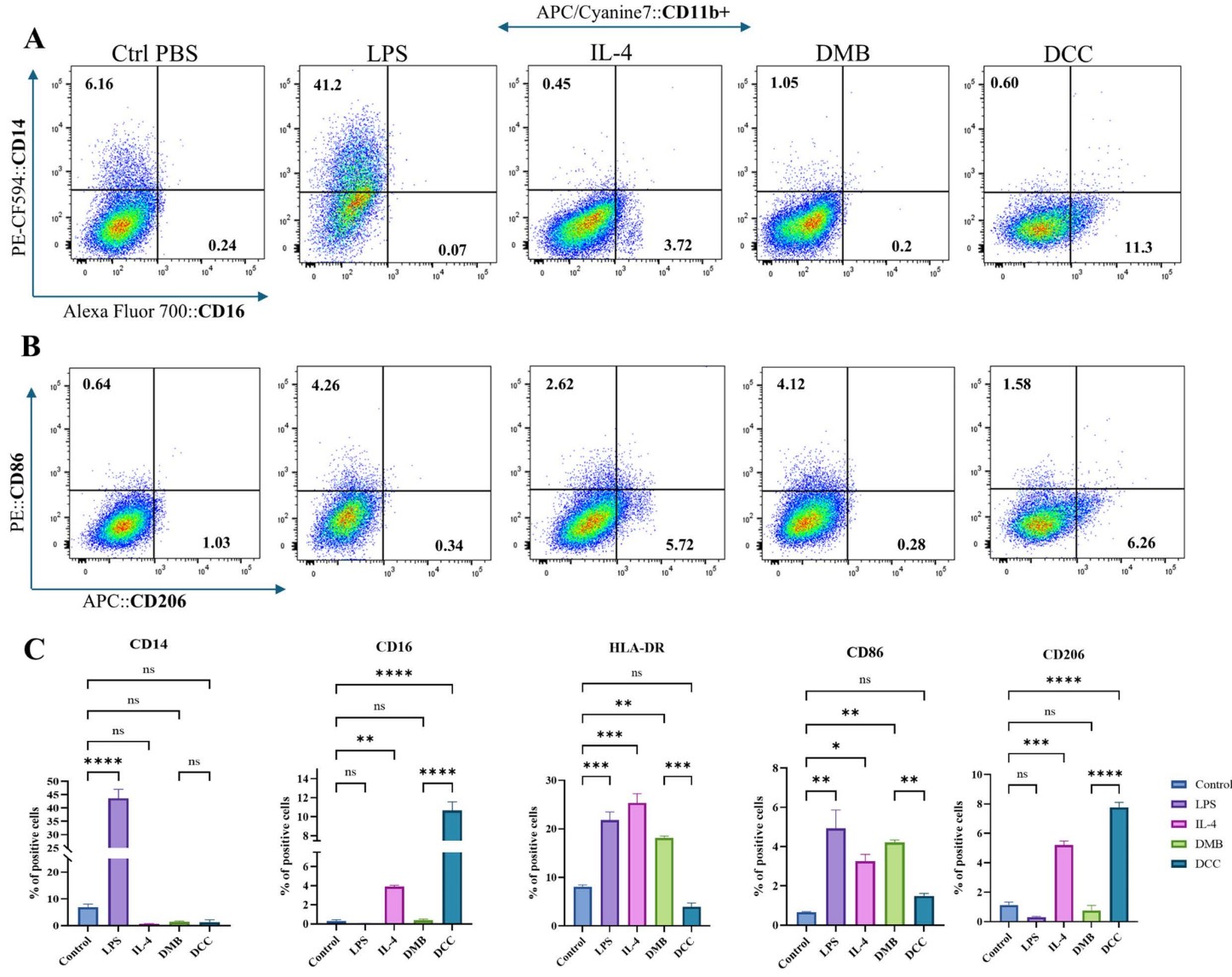

**Fig 9. Flow cytometry-based phenotypic profiling of macrophages treated for 72 hours with PBS, LPS, IL-4, DMB, and DCC.** The FACS plots in row (A) depict the differential distribution of pro-inflammatory versus anti-inflammatory subsets based on CD14 (Y-axis) and CD16 (X-axis) expression. The FACS plots in row (B) indicate macrophage polarization, as indicated by CD86 (Y-axis) and CD206 (X-axis) expression across all treatment groups. (C) presents a graphical representation of the expression of CD14, CD16, HLA-DR, CD86, and CD206 across all five groups. The notation ns indicates no significance, while ****p ≤ 0.0001, ***p ≤ 0.001, **p ≤ 0.01, and *p ≤ 0.05.

bone grafts, as well as their combined effects on the immunomodulatory responses of macrophages, to fully comprehend the FBR associated with these materials. This study examined the physicochemical properties of the DMB and DCC bone grafts while investigating the macrophage response to the adsorbed serum proteins on these grafts. We analysed the macrophage response to two distinct BAMPs profiles characterized by different physicochemical properties and adsorbed serum proteomic signatures. Our findings indicate that serum proteins adsorbed to the surfaces of bone grafts modulate inflammatory and anti-inflammatory responses in THP-1-derived macrophages, independent of direct interaction with the biomaterial.

  

DMB bone graft exhibited hydrophobic characteristics, with reduced anionic charge and increased surface roughness, leading to greater serum protein adsorption, as indicated by the micro-BCA assay. Conversely, the hydrophilic DCC, with lower surface roughness and higher anionic charge, resulted in reduced protein adsorption. These findings align with previous studies, which indicate that protein adsorption is higher on hydrophobic surfaces [37,38]. Protein adsorption generally occurs more easily on hydrophobic surfaces because fewer water molecules need to be displaced before adsorption. Hydrophilic surfaces tend to resist protein adsorption [39], and proteins tend to bond more strongly to hydrophilic surfaces [40]. Differences in the processing of two bone graft variations may have led to variations in surface properties and the associated adsorbed proteomic profiles.

Variations in surface properties between DMB and DCC resulted in distinct adsorption profiles of serum proteins. The hydrophobic characteristics and increased surface roughness of DMB may have led to higher concentrations of adsorbed serum proteins, with no significant downregulation of serum proteins when compared to 10% FBS, and significant upregulation of serum proteins such as tetranectin, cadherin 13, and C1q domain-containing protein. Conversely, the hydrophilic nature and higher anionic charge of DCC were associated with downregulation of serum proteins, including complement C9, vitronectin, fibrinogen, and fibronectin, as well as significant upregulation of serpin A3, also known as alpha-1-antichymotrypsin, on the DCC bone graft surface. At physiological pH, vitronectin, fibrinogen, and fibronectin carry a net negative charge [41–43], possibly resulting in significantly less adsorption on the anionic surface of DCC bone grafts. The electrostatic attraction between the protein and the biomaterial surface occurs through charge-charge interactions, with opposing charges favouring protein adsorption and like charges favouring reduced protein adsorption [44,45]. Extracellular matrix proteins, including cadherin 13 (T-cadherin), tetranectin, vitronectin, and fibronectin, contain hydrophobic residues [41,46,47] that may facilitate their adsorption onto the hydrophobic surface of DMB via hydrophobic interactions. Additionally, an increase in surface roughness increases the surface area available for protein adsorption [48,49], possibly resulting in increased protein adsorption on the surface of DMB. In contrast, DCC, characterized by its hydrophilic surface and high anionic charges, could have impeded the adsorption of these serum proteins.

Treatment of THP-1-derived macrophages with adsorbed serum proteins from DMB and DCC resulted in distinct phenotypic differentiation. In the DMB group, macrophages treated with DMSP displayed significant polarization towards an M1 pro-inflammatory phenotype, indicated by rounded morphology and increased expression of HLA-DR, CD86, and iNOS, as confirmed by FACS and confocal immunofluorescence, comparable to the positive inflammatory control LPS. Additionally, mRNA levels of IL-1β and TNF-α in the DMB group were elevated, correlating with increased cytokine concentrations in culture supernatants, as quantified by ELISA. Conversely, in the DCC group, macrophages treated with DCSP exhibited a spindle-shaped morphology and a higher proportion of M2 (anti-inflammatory) macrophages, as evidenced by increased expression of CD16, CD206, and Arg-1. These findings were confirmed by FACS and confocal immunofluorescence and were similar to the positive anti-inflammatory control (IL-4). The M2 predominance in the DCC group was further confirmed by the expression of anti-inflammatory cytokines IL-10 and TGF-β at both mRNA and protein levels, as determined using RT-PCR and ELISA. The findings confirm that the phenotypic modulation of macrophages on biomaterials occurs through the adsorption of serum proteins, which are influenced by the physicochemical properties of bone grafts, representing the central concept of BAMPs.

The increased presence of M2 macrophages in the DCC group and the predominance of M1 macrophages in the DMB group are consistent with our previous research [29], which primarily investigated macrophage polarization driven by treatment with DMB and DCC bone granules. Building on these findings, this study addresses an unexplored dimension by characterizing serum proteomic profiles adsorbed onto DMB and DCC surfaces and assessing their influence on macrophage phenotypic differentiation. By correlating serum protein adsorption with subsequent macrophage responses, this study provides a more comprehensive understanding of how BAMPs initiate early immune responses to bone grafts. This approach not only confirms our earlier findings but also enhances our understanding of how the substrate's surface

chemistry and adsorbed serum proteins interact to influence macrophage phenotypic differentiation within the tissue microenvironment.

Mass spectrometry proteomic profiling indicated that DMSP contained higher concentrations of fibrinogen, fibronectin, vitronectin, kallikrein, and complement factors, as demonstrated by heatmap and volcano plot analyses. These serum proteins have been reported to be associated with macrophage fusion and the formation of foreign body giant cells (FBGCs), which significantly influence the inflammatory response [50–52]. The increased adsorption of these proteins could have possibly contributed to a greater presence of pro-inflammatory macrophage subsets in the DMB group. In contrast, DCSP showed significant downregulation of these proteins, which could have reduced the M1 macrophage phenotype. Additionally, the DCSP exhibited higher levels of collagen type 1, serpin A3, alpha-1 antitrypsin, and arginase, which could be associated with anti-inflammatory effects and tissue repair [53–57]. The increased adsorption of these proteins might have resulted in a higher M2 macrophage population in the DCC group.

The present study identified distinct serum protein adsorption profiles on DMB and DCC bone graft surfaces, which were associated with differential macrophage polarization patterns. Although several adsorbed serum proteins on the surface of bone grafts have been linked to macrophage phenotypic polarization, the current study design does not establish a direct mechanistic conclusion between individual proteins and macrophage polarization outcomes. Macrophage polarization in DMB and DCC groups was governed by the collective influence of the adsorbed serum proteins, rather than by the action of a single dominant protein. The composition and possible conformational changes of the adsorbed serum protein layer, along with surface physicochemical properties such as roughness, charge, and wettability that govern the protein adsorption process, collectively influenced the macrophage polarization. Therefore, the polarization outcomes observed in the present study likely reflect the integrated effect of multiple adsorbed proteins and surface-associated cues, rather than single protein-specific mechanisms. The differential macrophage polarization observed in the DMB and DCC groups, associated with distinct protein adsorption profiles, represents a correlation rather than a confirmed mechanistic relationship, as no functional inhibition or gene-silencing experiments were conducted. Future studies employing functional validation approaches such as selective serum protein depletion, antibody-mediated blocking, receptor inhibition, or gene knockdown will be required to confirm the mechanistic roles of specific proteins in directing macrophage polarization.

Furthermore, it is essential to investigate the mechanisms underlying macrophages' phenotypic response to adsorbed serum proteins. These proteins exhibit a non-inflammatory response in interstitial fluid; however, adsorption onto a biomaterial's surface can cause protein unfolding, exposing epitopes that activate immune cells [12]. During adsorption, serum proteins spread to expose their hydrophobic cores, forming a monolayer and releasing water molecules bound to the native protein state [58]. This conformational change may enable the serum protein's RGD sequence to bind to macrophage integrins or toll-like receptors, thereby promoting phenotypic differentiation. Moreover, the results of this study underscore the need for further investigation into the macrophage receptors involved in the interactions with the RGD sequence of serum proteins.

Our findings also underscore the critical, yet underexplored, role of the immunomodulatory cytokine IL-10 in the early protein phase of the FBR and its subsequent osteogenic outcome. The significant association observed between DCC bone graft and elevated levels of IL-10 suggests that the smooth, hydrophilic biomaterial surfaces and their adsorbed serum protein profile display a BAMPs profile that appears to direct the immune response toward an anti-inflammatory, pro-regenerative phenotype. IL-10-type responses are of paramount importance, as they reflect and promote macrophage polarization towards an M2-like, anti-inflammatory state that dampens detrimental Th1 responses and thereby encourages bone growth and remodelling [59–61]. Therefore, it is possible that the adsorbed protein profile does not merely act as a passive scaffold but as a dynamic immunomodulatory interface. In other words, the induction of IL-10 may be a crucial mechanism by which the initial host-biomaterial interaction helps establish a microenvironment that supports bone healing. Should this be the case, the nature of the biomaterial used, and the profile of proteins that adsorb to it may represent active immune-guided regeneration tools rather than passive biocompatible moieties [62]. The switch of macrophage

polarization from a pro-inflammatory (M1) phenotype into an anti-inflammatory (M2) phenotype presents a novel opportunity to stimulate grafted bone tissue remodelling and healing [59]. In short, the development and use of biomaterials exhibiting a BAMPs profile that triggers M2 polarization and/or elicits IL-10-like responses to avoid excessive pathological inflammation following bone or biomaterial grafting, and possibly mitigate inflammation in bone graft patients suffering from the adverse consequences of FBR, are urgently needed.

Although this study provides an in-depth exploration of the BAMPs concept and highlights its significance in FBR, it acknowledges certain limitations. While providing valuable insights into the phenotypic modulation of macrophages based on adsorbed serum proteins, the approach may not fully capture the complexities of in vivo immune responses. Although we performed proteomic profiling of adsorbed serum proteins, we did not conduct functional assays to directly correlate specific proteins with macrophage polarization outcomes. The static protein adsorption model employed here may not accurately represent the dynamic protein exchange and remodelling present in physiological flow conditions within in vivo systems. Moreover, this study focused on the 72-hours response of macrophages, which are representative of the acute inflammatory phase of FBR; however, it did not address longer-term effects, such as osteogenic signalling, angiogenesis, or eventual bone regeneration.

## 5. Conclusion

This study demonstrated that the physicochemical characteristics of bone graft surfaces influence the proteomic profiles on DMB and DCC surfaces, thereby impacting macrophage phenotypic differentiation. Further work is needed to identify the key serum components associated with DMB and DCC bone grafts that modulate macrophage polarization. Moreover, the immunomodulatory effects of bone graft-associated BAMPs on other immune cell subsets require further exploration. A deeper understanding of these aspects of bone graft–immune response interactions will guide the design of biomaterials that mitigate FBR and promote graft vascularization and biological integration.

## Supporting information

**S1 Fig. The experimental workflow for protein adsorption and desorption on DMB and DCC bone grafts.** The total protein concentration in desorbed serum proteins was measured using the micro-BCA assay, and proteomic analysis was conducted to assess protein adsorption profile at one hour on the surfaces of DMB and DCC. (Fig. created using Biorender.com by Dr. Carel Brigi).
(TIF)

**S2 Fig. Experimental workflow to investigate the immunomodulatory effect of adsorbed serum proteins on macrophage phenotypic differentiation.** The cell treatment protocol for macrophages in the test and control groups, as well as the subsequent analysis for phenotypic differentiation assessment, is outlined. (Fig. created using Biorender.com by Dr. Carel Brigi).
(TIF)

**S3 Fig. Gating strategy for macrophage phenotypic differentiation.** Representative flow cytometry plots illustrate the sequential gating strategy employed for macrophage phenotypic differentiation. Initially, cells were gated based on SSC-A versus FSC-A to exclude debris. Subsequently, macrophages were identified by SSC-A versus CD11b expression. Finally, macrophage polarization was evaluated within the CD11b$^+$ population, with CD86 utilized to distinguish pro-inflammatory (M1-like) macrophages and CD206 to identify anti-inflammatory (M2-like) macrophages. Further characterization of macrophage subsets was performed within the CD11b$^+$ population based on CD14 and CD16 expression. All gating procedures were established with appropriate controls and applied consistently across all samples.
(TIF)

**S1 Table. Describes the antibodies and conjugated fluorochromes used for FACS analysis.**
(DOCX)

**S2 Table. Lists the ELISA kits used to detect cytokine concentrations in cell culture supernatants.**
(DOCX)

## Acknowledgments

The authors express their sincere gratitude to the Research Institute of Medical and Health Sciences (RIMHS) at the University of Sharjah for providing the necessary research facilities for this study. Special thanks are extended to Eng. Mohamed Shameer and Eng. Fahad Hassan of the Advanced Materials Research Laboratory at the University of Sharjah for their technical contributions to the SEM-EDS and WCA image analysis. Additionally, the authors recognize Mr. Manju Nidagodu Jayakumar for the technical support rendered in flow cytometric analysis and confocal microscopy imaging. The authors also acknowledge BioRender.com for the technical support in creating the figures used in the materials and methods section. The authors recognize that Grammarly (v1.2.177.1709) software was used to enhance readability and language, ensuring adherence to U.S. English standards.

## Author contributions

**Conceptualization:** Carel Brigi, Mawieh Hamad, AR Samsudin.

**Data curation:** Carel Brigi, Ensanya Ali Abou Neel, Balachandar Selvakumar, K. G. Aghila Rani, Sausan AlKawas, Hamza M. Al Hroub, Sherlyn Jemimah, Amin F. Majdalawieh, Amjad Mahasneh.

**Formal analysis:** Carel Brigi, Mawieh Hamad, Ensanya Ali Abou Neel, Balachandar Selvakumar, K. G. Aghila Rani, Sausan AlKawas, Hamza M. Al Hroub, Sherlyn Jemimah, Amin F. Majdalawieh, Amjad Mahasneh, AR Samsudin.

**Funding acquisition:** Balachandar Selvakumar, K. G. Aghila Rani, Sausan AlKawas, Amjad Mahasneh, AR Samsudin.

**Investigation:** Carel Brigi, Mawieh Hamad, Ensanya Ali Abou Neel, Balachandar Selvakumar, K. G. Aghila Rani, Sausan AlKawas, Hamza M. Al Hroub, Sherlyn Jemimah, Amin F. Majdalawieh.

**Methodology:** Carel Brigi, Mawieh Hamad, Ensanya Ali Abou Neel, Balachandar Selvakumar, K. G. Aghila Rani, Hamza M. Al Hroub, Sherlyn Jemimah, Amin F. Majdalawieh, Amjad Mahasneh, AR Samsudin.

**Project administration:** Carel Brigi, Mawieh Hamad, Ensanya Ali Abou Neel, Sausan AlKawas, Amjad Mahasneh, AR Samsudin.

**Resources:** Mawieh Hamad, Ensanya Ali Abou Neel, Balachandar Selvakumar, K. G. Aghila Rani, Sausan AlKawas, Hamza M. Al Hroub, Sherlyn Jemimah, Amin F. Majdalawieh, AR Samsudin.

**Software:** Carel Brigi, Mawieh Hamad, Ensanya Ali Abou Neel, K. G. Aghila Rani, Sausan AlKawas, Hamza M. Al Hroub, Sherlyn Jemimah, Amin F. Majdalawieh, Amjad Mahasneh, AR Samsudin.

**Supervision:** Carel Brigi, Mawieh Hamad, Ensanya Ali Abou Neel, Sausan AlKawas, Hamza M. Al Hroub, Sherlyn Jemimah, Amin F. Majdalawieh, AR Samsudin.

**Validation:** Carel Brigi, Mawieh Hamad, Ensanya Ali Abou Neel, Balachandar Selvakumar, K. G. Aghila Rani, Hamza M. Al Hroub, Sherlyn Jemimah, Amin F. Majdalawieh, Amjad Mahasneh, AR Samsudin.

**Visualization:** Carel Brigi, Mawieh Hamad, Ensanya Ali Abou Neel, Balachandar Selvakumar, K. G. Aghila Rani, Sausan AlKawas, Hamza M. Al Hroub, Sherlyn Jemimah, Amin F. Majdalawieh, Amjad Mahasneh, AR Samsudin.

**Writing – original draft:** Carel Brigi, Balachandar Selvakumar, K. G. Aghila Rani, Amjad Mahasneh.

**Writing – review & editing:** Carel Brigi, Mawieh Hamad, Ensanya Ali Abou Neel, Balachandar Selvakumar, K. G. Aghila Rani, Sausan AlKawas, Hamza M. Al Hroub, Sherlyn Jemimah, Amin F. Majdalawieh, Amjad Mahasneh, AR Samsudin.

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
