## [Decision Letter · Decision Letter 0]

10 Nov 2025

Dear Dr. Samsudin,

Thank you for submitting your manuscript to PLOS ONE. After careful consideration, we feel that it has merit but does not fully meet PLOS ONE’s publication criteria as it currently stands. Therefore, we invite you to submit a revised version of the manuscript that addresses the points raised by the reviewers during the review process.

We look forward to receiving your revised manuscript.

Kind regards,

Pradeep Kumar, Ph.D.

Academic Editor

PLOS ONE

PONE-D-25-46093

2. Please include a separate caption for each figure in your manuscript.

Additional Editor Comments (if provided):

Reviewers' comments:

Reviewer's Responses to Questions

**Comments to the Author**

1. Is the manuscript technically sound, and do the data support the conclusions?

Reviewer #1: Partly

Reviewer #2: Yes

Reviewer #3: Yes

Reviewer #4: Yes

2. Has the statistical analysis been performed appropriately and rigorously?

Reviewer #1: Yes

Reviewer #2: Yes

Reviewer #3: Yes

Reviewer #4: Yes

3. Have the authors made all data underlying the findings in their manuscript fully available?

Reviewer #1: Yes

Reviewer #2: No

Reviewer #3: Yes

Reviewer #4: Yes

4. Is the manuscript presented in an intelligible fashion and written in standard English?

Reviewer #1: No

Reviewer #2: Yes

Reviewer #3: Yes

Reviewer #4: Yes

Reviewer #1: This study investigated the physicochemical properties and adsorbed serum proteomic profiles of demineralized (DMB) and decellularized (DCC) bone grafts, as well as the immunomodulatory effects of serum proteins adsorbed onto these grafts on the phenotypic and functional profiles of macrophages. The work is well-conceived and supported by relevant literature. However, several issues need to be addressed before the manuscript can be considered for publication.

First, the manuscript contains grammatical errors and requires thorough revision to ensure consistency in the use of standard scientific English (e.g., coverslip vs. cover slips). Second, the description of critical methodological steps, particularly the protein adsorption and desorption processes from bone grafts and the THP-1 cell culture conditions, is insufficient and needs to be expanded for reproducibility.

In addition, the authors are encouraged to include correlation analyses for the evaluated markers to strengthen the interpretation of their results. For instance, assessing associations between LPS stimulation and iNOS production in DMB samples, or between IL-4 treatment and Arg-1 expression in DCC samples, would provide valuable mechanistic insight. The presentation of cytokine quantification should also be revised: the units (e.g., pg/mL) must be reported consistently with the values shown in Figure 6. Although IL-10 levels were lower compared to other cytokines, this molecule is a key immunoregulatory factor. The authors are strongly advised to explore the biological processes related to IL-10 and discuss its relevance in the context of their findings.

Reviewer #2: The manuscript describes an interesting and novel study, but it cannot be accepted in its current version. A minor revision is recommended considering the following comments:

1. The introduction must include the differences between decellularized and demineralized tissues, as well as a justification for their use in this study.

2. The approval of the ethics committee for conducting this study must be included in the manuscript.

3. In Section 2.2.3, why was SEM chosen for sample analysis if magnifications of 60x and 100x can be achieved with an optical microscope?

4. How were the time intervals chosen in Section 2.3.1?

5. In Section 2.4.2, are the DMSP and DCSP concentrations proposed by the authors or based on previous studies?

6. The method section must include references to the concentration and conditions evaluated if they are based on previous studies.

7. The references should be updated to the latest ones

Reviewer #3: The manuscript is well-written, with clear structure and logical flow. The abstract effectively summarizes the study, and the discussion contextualizes findings within existing literature. However the authors should consider revising the following:

1. The title is quite long and somewhat repetitive (e.g., "distinct" appears twice). The authors may consider this; "Biomaterial-Associated Molecular Patterns Modulate Macrophage Polarization in Bone Grafting"

Ensure consistent formatting of figures and tables (some legends are embedded in the text).

2. Show Scale bars in the images, what is there are not legible.

3. Clarify acronyms on first use (e.g., DMSP, DCSP).

Reviewer #4: Brigi et al present a study of absorbed proteins on bovine based bone grafts incubated with THP-1 cells and review the effects of the bone surface morphology and chemistry on CD14, CD16, CD86, CD206, IL-1beta, TBF-alpha, etc. Two forms of the bone substrate were investigated: demineralized bone graft (BMD) and decellularized bone graft (DCC). The authors strive to investigate the effect of the graft chemistry and morphology on the profile of absorbed proteins from FBS and their subsequent immunomodulatory effect on macrophages. Overall I find the presented work thorough and well devised. The statistical analysis is adequate. Based upon the submitted manuscript and the comments below, I recommend revision prior to publication.

(1) In section 2.1, the authors mention that the substrates undergo a sterilization process. No details of this process are given. Please include further details.

(2) In section 2.2.1, how were the samples prepared for water contact angle measurements? Were bone sections prepared? Or were the original surfaces used?

(3) In section 2.2.2, the bone samples were prepared into a fine powder. How was the bone powder prepared and to what nominal particle size for the zeta potential measurements?

(4) Based upon the work of the authors cited in Ref 30, I find this work to be a complement to the 2024 study. Unfortunately the work from e0300331 are not included in the discussion or introduction. The previous work focused on macrophage polarization but the present work strives to elucidate the substrate surface chemistry and protein absorption profile's effect on macrophage differentiation.

.

Reviewer #1: No

Reviewer #2: No

Reviewer #3: No

Reviewer #4: No

---

## [Author Response · Author response to Decision Letter 1]

16 Dec 2025

Academic Editor comments

Thank you for your comment. The manuscript has been prepared in accordance with the PLOS ONE style templates.

2. Please include a separate caption for each figure in your manuscript.

We appreciate your attention and thank you for informing us. A caption has been provided for each figure in accordance with the PLOS ONE style template.

We once again thank the academic editor for the important comment. As noted by Reviewer 4, the referenced publication has already been cited in our initial manuscript submission. Our previously published work complements the current study; therefore, we have incorporated several sentences discussing its relevance in the introduction and discussion sections, underscoring its connection to the present work.

Reviewer #1:This study investigated the physicochemical properties and adsorbed serum proteomic profiles of demineralized (DMB) and decellularized (DCC) bone grafts, as well as the immunomodulatory effects of serum proteins adsorbed onto these grafts on the phenotypic and functional profiles of macrophages. The work is well-conceived and supported by relevant literature. However, several issues need to be addressed before the manuscript can be considered for publication.

Thank you for your time and effort in reviewing our study. We appreciate your comments on the data presentation, and we have revised it accordingly. We also thank you for your remarks on our research, which you found well-conceived and supported by relevant literature.

1. First, the manuscript contains grammatical errors and requires thorough revision to ensure consistency in the use of standard scientific English (e.g., coverslip vs. cover slips).

We appreciate the reviewer for highlighting this point. The manuscript has been thoroughly revised and proofread multiple times. Grammarly (v1.2.177.1709) was used to enhance readability and language, ensuring adherence to U.S. English standards.

2. Second, the description of critical methodological steps, particularly the protein adsorption and desorption processes from bone grafts and the THP-1 cell culture conditions, is insufficient and needs to be expanded for reproducibility.

We understand the importance of this comment. Therefore, we have included detailed methodological steps for protein adsorption and desorption, as well as for cell treatment. Additionally, we have included detailed supplementary figures illustrating the protein adsorption and desorption process and cell treatment methodology in the manuscript.

Please find the details in Materials and Methods

Section: 2.3.1, Pg: 6-8, Line: 162-194.

Section: 2.4.3 Pg: 10, Line: 240-254.

Figures: S1 Fig, S2 Fig

3. In addition, the authors are encouraged to include correlation analyses for the evaluated markers to strengthen the interpretation of their results. For instance, assessing associations between LPS stimulation and iNOS production in DMB samples, or between IL-4 treatment and Arg-1 expression in DCC samples, would provide valuable mechanistic insight.

We appreciate the reviewer's suggestion to include additional statistical analyses. However, in this study, the methods used and data collected were based on a single concentration and time point. As a result, the study design does not support robust correlation analysis, which generally requires experimental conditions with different concentrations or time points to evaluate associations accurately. Consequently, correlation analysis was not conducted in this work. We plan to include experimental designs with varying concentrations in future studies to facilitate comprehensive correlation analysis.

4. The presentation of cytokine quantification should also be revised: the units (e.g., pg/mL) must be reported consistently with the values shown in Figure 6.

Figure 6 has been renumbered as Figure 7.

We would like to draw your kind attention to the details in Figure 7. The upper set of histograms, from (A) to (D), shows mRNA expression of cytokines relative to GAPDH. Meanwhile, the lower sets of histograms, from (E) to (H), show the protein expression in pg/ml. We made a slight modification to this figure by adding a line between the two sets and including a small heading to improve visibility, indicating that the upper histograms represent mRNA expression and the lower histograms show protein concentration (pg/mL). Please find the revised Figure 7.

5. Although IL-10 levels were lower compared to other cytokines, this molecule is a key immunoregulatory factor. The authors are strongly advised to explore the biological processes related to IL-10 and discuss its relevance in the context of their findings.

We appreciate the reviewer's insightful comment. Indeed, IL-10 is a critical immunoregulatory cytokine, and its role in bone graft integration, as described by our data, deserves further discussion. That said, and not to overstate the significance of IL-10 in bone grafting and regeneration, we included a brief paragraph further discussing the broader relevance of IL-10 in relation to our findings.

Please see the discussion section.

Section: 4, Pg: 25, Lines 627-645

Reviewer #2: The manuscript describes an interesting and novel study, but it cannot be accepted in its current version. A minor revision is recommended, considering the following comments:

We appreciate the reviewer's comments and are pleased that the reviewer found the study both innovative and interesting. We also thank you for your suggestion of minor revisions.

1. The introduction must include the differences between decellularized and demineralized tissues, as well as a justification for their use in this study.

Thank you for pointing this out to us. We have added a brief paragraph outlining the differences between decellularized and demineralized tissues, along with a justification for their use in this study.

Please see the introduction section,

Section: 1, Pg: 4, Lines: 95-106

2. The approval of the ethics committee for conducting this study must be included in the manuscript

We used bovine bone samples obtained from a certified slaughterhouse, and the THP-1 cell line was acquired as described in the methodology. Based on our materials and methods, ethical approval from the ethics committee is not required. We also acknowledge the importance of this comment. Therefore, we are providing a letter stating that this study does not necessitate ethical approval from the University of Sharjah Animal Care and Use Committee.

Please find the attached file titled Ethical Committee Letter.

3. In Section 2.2.3, why was SEM chosen for sample analysis if magnifications of 60x and 100x can be achieved with an optical microscope?

We appreciate the reviewer's comment on the details of the results. While magnifications of 60× and 100× are achievable with standard optical microscopy, we chose Scanning Electron Microscopy (SEM) for its superior surface clarity and structural resolution. Optical microscopy was initially insufficient for clearly delineating the scaffold architecture needed for accurate roughness measurement; in contrast, SEM provided enhanced depth of field, contrast, and surface detail in high-quality micrographs. As a result, SEM images at 60× and 100× magnifications were deemed more reliable for assessing surface roughness using ImageJ. We previously mentioned in the initial manuscript submission that SEM images were used for surface roughness analysis in ImageJ software.

Please see the materials and methods section,

Section: 2.2.3, Pg: 6, Lines: 158-159

4. How were the time intervals chosen in Section 2.3.1?

The time interval was chosen based on a previously published article about time-dependent protein adsorption on surfaces. However, the initial and final time points corresponded to the early phase of protein adsorption relevant to the FBR process in bone grafts. The time interval reference is provided in the methodology section.

Please see the Materials and Methods section.

Section: 2.3.1, Pg: 7, Lines: 166-173

5. In Section 2.4.2, are the DMSP and DCSP concentrations proposed by the authors or based on previous studies?

Thank you for highlighting this point. We have included the reference from the previous study that explains the treatment concentration of the extracellular matrix protein.

Please see the methodology section,

Section: 2.4.2, Pg:9, Lines: 233-234

6. The method section must include references to the concentration and conditions evaluated if they are based on previous studies.

Thank you for informing us. We have established the concentrations and conditions based on our experimental results. The concentrations were chosen based on our XTT assay results.

Section: 3.6, Pg: 18, Lines 441-444

7. The references should be updated to the latest ones

Thank you for bringing it to our attention. We have revised the reference accordingly.

Please find the reference section. Pg 27-34

Reviewer #3: The manuscript is well-written, with clear structure and logical flow. The abstract effectively summarizes the study, and the discussion contextualizes findings within existing literature. However, the authors should consider revising the following:

We appreciate the reviewer's efforts and the time dedicated to evaluating the manuscript. We also acknowledge your comments that the manuscript is clear, well-structured, and logical in its presentation.

1. The title is quite long and somewhat repetitive (e.g., "distinct" appears twice). The authors may consider this; "Biomaterial-Associated Molecular Patterns Modulate Macrophage Polarization in Bone Grafting." Ensure consistent formatting of figures and tables (some legends are embedded in the text).

Thank you for suggesting the new title. We have revised it accordingly and are pleased to implement the proposed change. Additionally, we have informed the academic editor about this update.

Thank you for informing us. We have amended and reformatted the figures and tables in accordance with the PLOS ONE style template.

2. Show Scale bars in the images, what is there are not legible.

Thank you for pointing this out to us. We have improved the clarity of the figure, so the scales are more visible. Please see the figures attached.

3. Clarify acronyms on first use (e.g., DMSP, DCSP).

Thank you. We have corrected it accordingly.

Please see the materials and methods section,

Section: 2.3.1, Pg: 8, Line 191

Reviewer #4: Brigi et al present a study of absorbed proteins on bovine based bone grafts incubated with THP-1 cells and review the effects of the bone surface morphology and chemistry on CD14, CD16, CD86, CD206, IL-1beta, TBF-alpha, etc. Two forms of the bone substrate were investigated: demineralized bone graft (BMD) and decellularized bone graft (DCC). The authors strive to investigate the effect of the graft chemistry and morphology on the profile of absorbed proteins from FBS and their subsequent immunomodulatory effect on macrophages. Overall I find the presented work thorough and well devised. The statistical analysis is adequate. Based upon the submitted manuscript and the comments below, I recommend revision prior to publication.

We thank the reviewer for the comments that could improve the detail of this manuscript. We also appreciate the positive comments recognizing the work as thorough and well devised, and for recommending revision before publication.

1. In section 2.1, the authors mention that the substrates undergo a sterilization process. No details of this process are given. Please include further details.

Thank you for bringing that to our attention. We have included the details of the sterilization process in section 2.1.

Please see the materials and methods section.

Section: 2.1, Pg: 5, Lines 124-125

2. In section 2.2.1, how were the samples prepared for water contact angle measurements? Were bone sections prepared? Or were the original surfaces used?

Bone samples were not sectioned, and the original surface was used for the contact angle measurements. We have included a statement regarding this.

Please see the materials and methods section.

Section: 2.2.1, Pg: 5, Lines 130-131

3. In section 2.2.2, the bone samples were prepared into a fine powder. How was the bone powder prepared, and to what nominal particle size for the zeta potential measurements?

Thank you for bringing this to our attention. These details are included in the methodology section, where the bone sample preparation, method, and instrument used are specified. Additionally, the nominal particle size for zeta potential is provided in the results section.

Please see the materials and methods section.

Section: 2.2.2, Pg: 5,6, Lines 138-145

Please see the result section.

Section: 3.2, Pg: 14, Lines 338-339

4. Based upon the work of the authors cited in Ref 30, I find this work to be a complement to the 2024 study. Unfortunately the work from e0300331 are not included in the discussion or introduction. The previous work focused on macrophage polarization but the present work strives to elucidate the substrate surface chemistry and protein absorption profile's effect on macrophage differentiation.

We thank the reviewer for this insightful and in-depth consideration of the exact scope of our study. We briefly describe, in the discussion and introduction, how the present study complements our previous work.

Please see the introduction section.

Section: 1, Pg: 4, Lines 107-111

Please see the discussion section.

Section: 4, Pg: 24, Lines 598-607

---

## [Decision Letter · Decision Letter 1]

12 Jan 2026

Dear Dr. Samsudin,

Thank you for submitting your manuscript to PLOS ONE. After careful consideration, we feel that it has merit but does not fully meet PLOS ONE’s publication criteria as it currently stands. Therefore, we invite you to submit a revised version of the manuscript that addresses the points raised by Reviewer 3 during the review process.

We look forward to receiving your revised manuscript.

Kind regards,

Pradeep Kumar, Ph.D.

Academic Editor

PLOS One

Journal Requirements:

Reviewers' comments:

Reviewer's Responses to Questions

**Comments to the Author**

Reviewer #2: All comments have been addressed

Reviewer #3: All comments have been addressed

Reviewer #4: All comments have been addressed

2. Is the manuscript technically sound, and do the data support the conclusions?

Reviewer #2: Yes

Reviewer #3: Yes

Reviewer #4: Yes

3. Has the statistical analysis been performed appropriately and rigorously?

Reviewer #2: Yes

Reviewer #3: Yes

Reviewer #4: Yes

4. Have the authors made all data underlying the findings in their manuscript fully available?

Reviewer #2: Yes

Reviewer #3: Yes

Reviewer #4: Yes

5. Is the manuscript presented in an intelligible fashion and written in standard English?

Reviewer #2: Yes

Reviewer #3: Yes

Reviewer #4: Yes

Reviewer #2: The authors have improved the manuscript by considering the comments of the reviewers. The manuscript muss be accepted for publication in its revised version.

Reviewer #3: 1.Ensure consistent tense and terminology (e.g., “cover slip” vs. “coverslip”)

2.Consider shortening the abstract to meet word limit and improve readability

3. Check resolution and legibility of SEM images and scale bars

Reviewer #4: The authors have addressed my previous comments in this version of the manuscript. I appreciate the responses of the authors. I recommend publication as is.

.

Reviewer #2: No

Reviewer #3: No

Reviewer #4: No

---

## [Author Response · Author response to Decision Letter 2]

2 Feb 2026

Journal Requirements

We thank the editorial board for reminding us of this important point. As no specific references were recommended by the reviewer, no changes to the reference list were made in this revision. All references suggested in the earlier revision were thoroughly assessed for relevance, and only those pertinent to the objectives of the present study were incorporated.

The reference list was carefully reviewed for completeness, accuracy, and retraction status. To the best of our knowledge, none of the cited references have been retracted to date; therefore, no changes to the reference list were required. References have been formatted in accordance with the PLOS ONE guidelines.

Reviewer #3

1. Ensure consistent tense and terminology (e.g., “cover slip” vs. “coverslip”)

We thank the reviewer for this comment. The manuscript has been revised to ensure consistency in tense and terminology, and the term “coverslip” is now used consistently throughout the text.

Line: 257, 260, 265, 267, 455

2. Consider shortening the abstract to meet the word limit and improve readability.

We thank the reviewer for this comment. The abstract has been condensed to comply with the PLOS ONE word limit and to enhance readability.

Section: Abstract, Pg: 2, Line: 36-57.

3. Check resolution and legibility of SEM images and scale bars

We thank the reviewer for this comment. SEM images (Fig. 2 & Fig. 6) and figures containing scale bars were enlarged and converted to 600 dpi. All figures were prepared using PLOS’s free figure tool (NAAS) to ensure compliance with PLOS ONE publication-quality figure guidelines.

---

## [Decision Letter · Decision Letter 2]

16 Feb 2026

Dear Dr. Samsudin,

Thank you for submitting your manuscript to PLOS ONE. After careful consideration, we feel that it has merit but does not fully meet PLOS ONE’s publication criteria as it currently stands. Therefore, we invite you to submit a revised version of the manuscript that addresses the points raised by Reviewers 3 and 4 during the review process.

We look forward to receiving your revised manuscript.

Kind regards,

Pradeep Kumar, Ph.D.

Academic Editor

PLOS One

Journal Requirements:

Reviewers' comments:

Reviewer's Responses to Questions

**Comments to the Author**

Reviewer #2: All comments have been addressed

Reviewer #3: All comments have been addressed

Reviewer #4: (No Response)

2. Is the manuscript technically sound, and do the data support the conclusions?

Reviewer #2: Yes

Reviewer #3: Yes

Reviewer #4: Yes

3. Has the statistical analysis been performed appropriately and rigorously?

Reviewer #2: Yes

Reviewer #3: Yes

Reviewer #4: Yes

4. Have the authors made all data underlying the findings in their manuscript fully available?

Reviewer #2: Yes

Reviewer #3: Yes

Reviewer #4: Yes

5. Is the manuscript presented in an intelligible fashion and written in standard English?

Reviewer #2: Yes

Reviewer #3: Yes

Reviewer #4: Yes

Reviewer #2: The authors improved the manuscript by considering the comments of the reviewers. The manuscript must be accepted for publication in its revised version.

Reviewer #3: Interpretation of Causality Requires Caution

At several points, the manuscript implies direct mechanistic links between individual proteins and macrophage outcomes (e.g., fibrinogen ↔ M1 polarity; serpin A3 ↔ M2 phenotype). These interpretations are plausible but not demonstrated experimentally. Recommendation:

Clarify that these are correlations, not mechanistic conclusions, unless further functional inhibition/knockdown assays are added.

Reviewer #4: The authors have attempted to address Reviewer 3's comments regarding the SEM images. However, I find that he scale bars are not legible on the images. It may be necessary to at least state in the legend the value of the scale bar for each image. At 600 dpi the scale bar legends cannot be read.

Therefore, I would recommend revision of Figures 2 and 6 and the corresponding captions.

.

Reviewer #2: No

Reviewer #3: No

Reviewer #4: No

---

## [Author Response · Author response to Decision Letter 3]

19 Feb 2026

Journal Requirements

We thank the editorial board for reminding us of this important point. As no specific references were recommended by the reviewer, no changes to the reference list were made in this revision. All references suggested in the earlier revision were thoroughly assessed for relevance, and only those pertinent to the objectives of the present study were incorporated.

We thank the editorial board for reminding us of this point. The reference list was carefully reviewed for completeness, accuracy, and retraction status. To the best of our knowledge, none of the cited references have been retracted to date; therefore, no changes to the reference list were required. References have been formatted in accordance with the PLOS ONE guidelines.

Reviewer # 3

Interpretation of Causality Requires Caution. At several points, the manuscript implies direct mechanistic links between individual proteins and macrophage outcomes (e.g., fibrinogen ↔ M1 polarity; serpin A3 ↔ M2 phenotype). These interpretations are plausible but not demonstrated experimentally. Recommendation: Clarify that these are correlations, not mechanistic conclusions, unless further functional inhibition/knockdown assays are added.

Thank you very much for highlighting our interpretation of the results. We fully agree that our interpretation is merely correlational and not a mechanistic conclusion unless future studies are conducted. Therefore, we have rephrased the text in the respective paragraph to avoid misinterpretation of our claims as follows:

Discussion section; Page: 24,25; Lines: 609-636.

Reviewer # 4

1. The authors have attempted to address Reviewer 3's comments regarding the SEM images. However, I find that the scale bars are not legible on the images. It may be necessary to at least state in the legend the value of the scale bar for each image. At 600 dpi the scale bar legends cannot be read. Therefore, I would recommend revision of Figures 2 and 6 and the corresponding captions.

We thank the reviewer for this comment regarding the SEM images (Figures 2 and 6). To preserve image integrity and avoid selective enlargement that could potentially alter visual interpretation, the scale bar values for each panel have now been clearly specified in the revised figure legends. As recommended, the scale bar measurements are explicitly stated for each image, and the figures have been maintained at 600 dpi resolution in accordance with journal guidelines.

Result Section; Page: 15; Line: 357-360

Results Section; Page: 18; Line: 457-458

All figures were prepared using PLOS’s free figure tool (NAAS) to ensure compliance with PLOS ONE publication-quality figure guidelines.

---

## [Decision Letter · Decision Letter 3]

10 Mar 2026

Biomaterial-Associated Molecular Patterns (BAMPs) Modulate Macrophage Polarization in Bone Grafting

PONE-D-25-46093R3

Dear Dr. Samsudin,

We’re pleased to inform you that your manuscript has been judged scientifically suitable for publication and will be formally accepted for publication once it meets all outstanding technical requirements.

Kind regards,

Pradeep Kumar, Ph.D.

Academic Editor

PLOS One

Additional Editor Comments (optional):

Reviewers' comments:

Reviewer's Responses to Questions

**Comments to the Author**

Reviewer #3: All comments have been addressed

Reviewer #4: All comments have been addressed

2. Is the manuscript technically sound, and do the data support the conclusions?

Reviewer #3: Yes

Reviewer #4: Yes

3. Has the statistical analysis been performed appropriately and rigorously?

Reviewer #3: Yes

Reviewer #4: Yes

4. Have the authors made all data underlying the findings in their manuscript fully available?

Reviewer #3: Yes

Reviewer #4: Yes

5. Is the manuscript presented in an intelligible fashion and written in standard English?

Reviewer #3: Yes

Reviewer #4: Yes

Reviewer #3: The authors have addressed all the concerns raised, and the manuscript is in an acceptable form. Good work done.

Reviewer #4: The authors have addressed my previous comments. I recommend that the article be accepted in it's present form for publication.

.

Reviewer #3: No

Reviewer #4: No

---

## [Editor Report · Acceptance letter]

PONE-D-25-46093R3

PLOS One

Dear Dr. Samsudin,

I'm pleased to inform you that your manuscript has been deemed suitable for publication in PLOS One. Congratulations! Your manuscript is now being handed over to our production team.

Kind regards,

on behalf of

Prof. Pradeep Kumar

Academic Editor

PLOS One